# Integrated omics dissection of proteome dynamics during cardiac remodeling

Edward Lau[1,2,7], Quan Cao [1,2,3], Maggie P.Y. Lam[1,2,8], Jie Wang[1,2], Dominic C.M. Ng[1,2], Brian J. Bleakley [1,2], Jessica M. Lee[1,2], David A. Liem[1,2], Ding Wang[1,2], Henning Hermjakob [1,4] & Peipei Ping[1,2,5,6]

Transcript abundance and protein abundance show modest correlation in many biological models, but how this impacts disease signature discovery in omics experiments is rarely explored. Here we report an integrated omics approach, incorporating measurements of transcript abundance, protein abundance, and protein turnover to map the landscape of proteome remodeling in a mouse model of pathological cardiac hypertrophy. Analyzing the hypertrophy signatures that are reproducibly discovered from each omics data type across six genetic strains of mice, we find that the integration of transcript abundance, protein abundance, and protein turnover data leads to 75% gain in discovered disease gene candidates. Moreover, the inclusion of protein turnover measurements allows discovery of post-transcriptional regulations across diverse pathways, and implicates distinct disease proteins not found in steady-state transcript and protein abundance data. Our results suggest that multi-omics investigations of proteome dynamics provide important insights into disease pathogenesis in vivo.

[1] NIH BD2K Center of Excellence in Biomedical Computing, Los Angeles, CA 90095, USA. [2] Department of Physiology, David Geffen School of Medicine at UCLA, Los Angeles, CA 90095, USA. [3] Shanghai Institute of Cardiovascular Diseases, Zhongshan Hosptial, Fudan University, Shanghai 200032, China. [4] Molecular Systems Cluster, European Molecular Biology Laboratory-European Bioinformatics Institute, Wellcome Genome Campus, Cambridge CB10 1SD, UK. [5] Department of Medicine, David Geffen School of Medicine at UCLA, Los Angeles, CA 90095, USA. [6] Department of Bioinformatics, David Geffen School of Medicine at UCLA, Los Angeles, CA 90095, USA. [7] Present address: Stanford Cardiovascular Institute, Stanford University, Stanford, CA 94305, USA. [8] Present address: Department of Medicine, University of Colorado Anschutz Medical Campus, Aurora, CO 80045, USA. Correspondence and requests for materials should be addressed to P.P. (email: pping38@g.ucla.edu)

During pathological cardiac hypertrophy, the expression profile of cardiac proteins changes in response to discrete transcriptional regulation pathways[1–3] and global proteostatic perturbations[4,5]. Both transcriptomics (e.g., RNA-seq) and proteomics (e.g., mass spectrometry) experiments are now routinely employed to identify disease signatures and pathogenic mechanisms in cardiac hypertrophy and other diseases. It is becoming clear in many systems however that the differential expression of transcripts correlates poorly with that of proteins[6–12], prompting discussions on whether transcriptomics and proteomics experiments reflect the same biological regulations. Moreover, it is debated whether transcript-protein non-correlation reflects the dominance of distinct translational regulations on protein abundance[13–16], or instead results from unaccounted technical variability[6,11,17,18]. Although the relative contributions of transcriptional and translational processes are likely to be specific to the particular system under investigation, this debate has practical implications for experimental designs and resource allocation in biomedical studies that aim to capture representative molecular information of disease states for hypothesis generation or biomarker discovery. In other words, if transcript and protein abundance measure identical biological processes, resources may arguably be more effectively allocated for repeating transcriptomics experiments rather than performing proteomics analyses, or vice versa. Despite its important implications, how transcript-protein non-correlation impacts disease signature discovery in biomedical research has not been adequately evaluated.

Here we report an integrated analysis of transcriptomics, proteomics, and proteome dynamics data to examine their combined contribution to disease gene discovery in a pathological model of cardiac hypertrophy in inbred mice from six diverse genetic backgrounds. Protein turnover alters broadly in response to physiological and pathological perturbations[19–21], suggesting it may provide additional insights into the total landscape of proteome regulation in vivo. Thus far, measurements of turnover rate constants that provide direct evidence of temporal dynamics variations have been largely limited to cultured cells, whereas measurements of protein turnover rates in vivo remain uncommon due to technical challenges. The disease relevance of protein turnover in vitro is complicated by the dissimilar temporal regimes of protein degradation between cultured cells and whole-animal physiology (half-life of hours vs. weeks)[22,23]. The longer lifetime of proteins in vivo can therefore buffer transient transcript changes, and create additional demands on maintaining protein quality over long timespans[6,22,24]. Cardiac hypertrophy provides an excellent model to examine protein expression under multifaceted transcriptional and post-transcriptional influences. Comparing confident disease signatures that are consistently identified across six mouse strains, we observe that protein turnover, protein abundance, and transcript abundance nominate nonredundant candidate signatures of cardiac hypertrophy. Moreover, variations in protein turnover are independently associated with protein functional association and cardiac disease phenotype, suggesting a potential mechanism that uncouples various omics measurements. Altogether, the multi-omics data sets nominate 273 candidate disease signatures in 36 non-redundant cardiac pathways to be reproducibly altered in pathological cardiac hypertrophy, including 70 signatures and 13 pathways discovered only after the incorporation of protein turnover measurements.

## Results

**Multi-omics discovery of disease signatures in hypertrophy.** To quantify the abundance and turnover rates of cardiac proteins, we previously developed a strategy that combines in vivo metabolic labeling, high-resolution mass spectrometry, and computational kinetic modeling[21,25] (Fig. 1a). Animals were labeled with isotopes in vivo via deuterium oxide administration in drinking water for up to 14 days. Newly synthesized proteins become labeled with deuterium, whereas degradation of a labeled protein removes deuterium from the protein pool. The changes in proportion of isotopes in the protein pool of each protein species were then measured at 7 time points using a high-resolution Orbitrap mass spectrometer. We developed a computational software, ProTurn, which integrates mass isotopomer signals of all peptides at multiple time points, and utilizes a custom two-compartment kinetics model to calculate protein turnover rates $(k)$[25]. We define $k$ here empirically as the measured rate (per day) of label incorporation proportional to the protein pool. Because the protein pool size can change under physiological or pathological growth as well as differential gene expression, the measured parameter is a function of hidden temporal processes including protein synthesis and degradation. To model the variability of protein expression across individual animals[26], we performed experiments in six genetically diverse inbred strains (C57BL/6J, DBA/2J, CE/J, A/J, FVB/NJ, and BALB/cJ; seven time points each) to capture variations of protein turnover rates across healthy genetic backgrounds as well as their varying responses to chronic isoproterenol stimulation, a well-established experimental model of pathological cardiac hypertrophy (Supplementary Table 1)[27,28].

In total, we completed 1404 mass spectrometry experiments, identifying 8064 proteins and quantifying the turnover and abundance of 3228 proteins from 120,454 peptide time-series (Supplementary Fig. 1)[25]. Comparison of the acquired protein abundance profile against a public data set on the human proteome[29] confirms that protein expression closely resembles the adult heart out of 30 cell/tissue types, confirming our mass spectrometry protein quantification specifically captures cardiac protein expression profiles (Supplementary Fig. 1). Our data show that cardiac proteins are continually replaced in vivo at a median half-life of 7.7 days. A substantial variation of turnover rates exists amongst quantified proteins, with the 5th–95th percentile range spanning >10-fold (half-life 1.4–19.7 days), consistent with post-translational regulatory mechanisms capable of regulating protein steady-state abundance. We further integrated the data set with an existing cardiac transcript expression data set (GSE48760). The data were obtained from identical mouse strains under isoproterenol challenge in an independent study[27]. A subset of 1901 genes/proteins that are commonly shared in all six strains has been selected for analysis, as these 1901 proteins possessed quantified transcript abundance, protein abundance, and protein turnover data across the six strains. As expected, we observed poor correlations between changes in protein turnover in hypertrophy with changes in protein abundance ($\rho = 0.11$) (Fig. 1b), changes in transcript abundance ($\rho = 0.09$) (Fig. 1c), or basal turnover flux ($\rho = -0.17$) (Supplementary Fig. 2).

To evaluate how this impacts the discovery of hypertrophy disease genes and/or proteins (disease signatures), we asked which genes/proteins are consistently identified across mouse strains in each of the three parameters (transcript abundance, protein abundance, and protein turnover), and which are thus more likely to be discovered as reproducible hypertrophy signatures in omics experiments. We define a putative consistent disease signature here as a gene/protein exhibiting changes in 80th or above percentile in ranks in four or more genetic backgrounds, followed by a combined test for significance. From the transcript abundance, protein abundance, and protein turnover data, a total of 273 genes/proteins are implicated in cardiac hypertrophy in four or more mouse strains (Fig. 1d). They include 49 genes/proteins with consistently increased

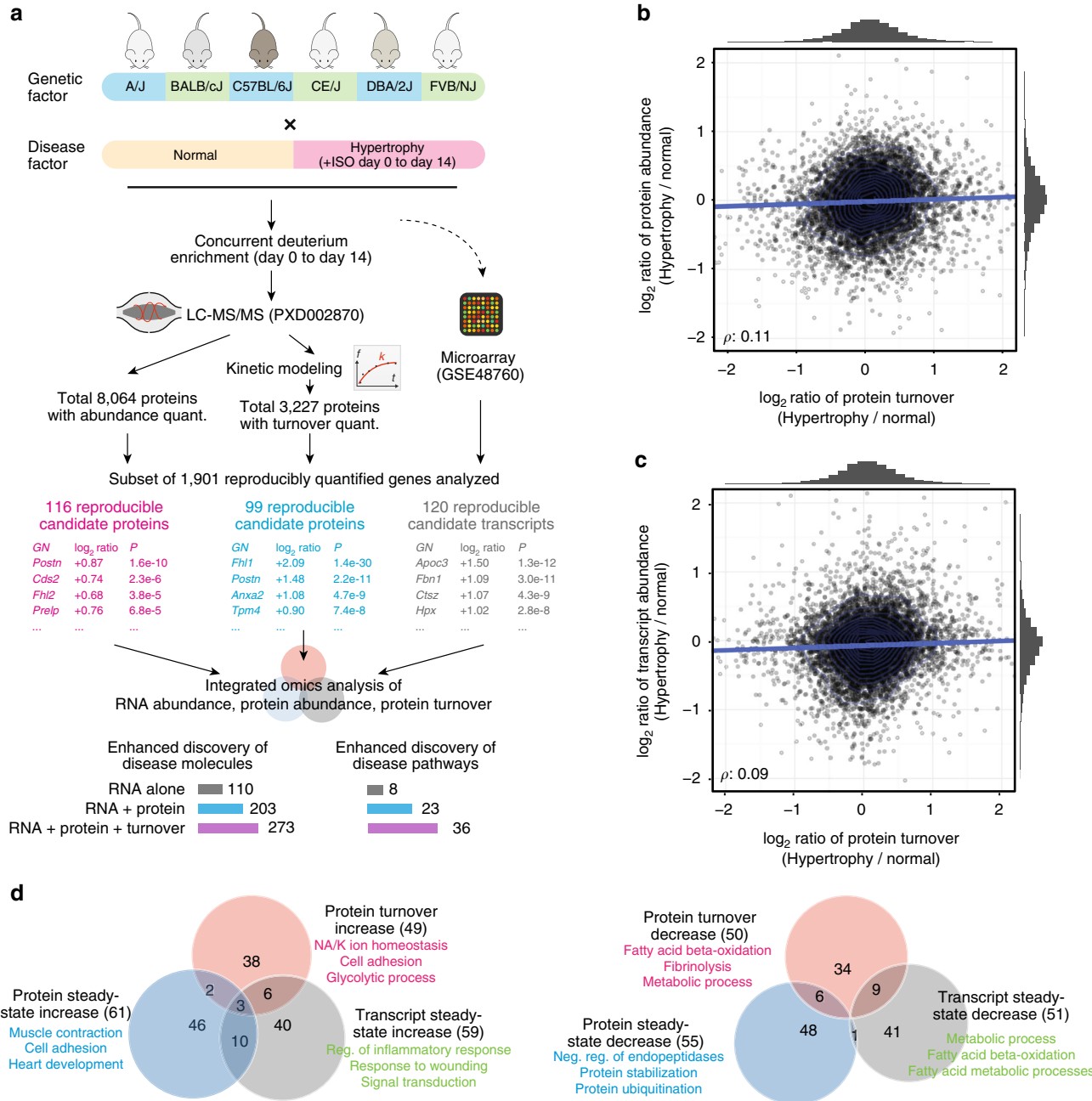

**Fig. 1** Signatures of cardiac hypertrophy at transcript, protein, and turnover levels. **a** Schematic of multi-omics analysis. **b** Plot of log ratios of protein turnover in hypertrophic vs. normal hearts (x-axis) against log ratios of protein abundance (y-axis) across six mouse strains (n = 5593 data points). Changes in protein turnover are poorly correlated with changes in protein abundance, accounting for only 11% of variations in protein abundance. Blue line: linear regression; rug: data density; rho: Spearman's correlation coefficient. **c** Plot of log ratios of protein turnover in hypertrophic vs. normal hearts (x-axis) against log ratios transcript abundance ratio (y-axis) (NCBI GEO GSE48760) (n = 5601 data points). Changes in protein turnover are minimally correlated with changes in transcript abundance, accounting for only 9% of variations in protein abundance. Blue line: linear regression; rug: data density; rho: Spearman's correlation coefficient. Steady-state and turnover changes to isoproterenol are largely orthogonal. **d** Venn diagram of consistently implicated candidate disease signatures identified at the transcript, protein and protein turnover levels. Candidate disease markers are defined as having shown increases (left) or decreases (right) at 80th percentile or above in ≥4 mouse strain. Enriched Gene Ontology (GO) biological processes in genes with differential protein turnover, protein steady-state abundance are also shown

protein turnover and 50 with decreased protein turnover; 61 with increased protein abundance and 55 with decreased protein abundance; 59 with increased transcript abundance and 51 with decreased transcript abundance. We reason that if disease signatures in hypertrophy are primarily driven by transcript-level changes, similarity between reproducible disease signatures at transcript and protein levels would be expected. Instead, we

observed low commonality in candidate disease proteins when comparing turnover disease signatures with protein abundance signatures (12%) or transcript abundance signatures (18%) (Fig. 1d). This observation persisted when an alternative method of protein quantification using tandem mass tag was performed (Supplementary Fig. 3). From the 273 candidate disease signatures, 99 proteins exhibit consistently altered turnover rates;

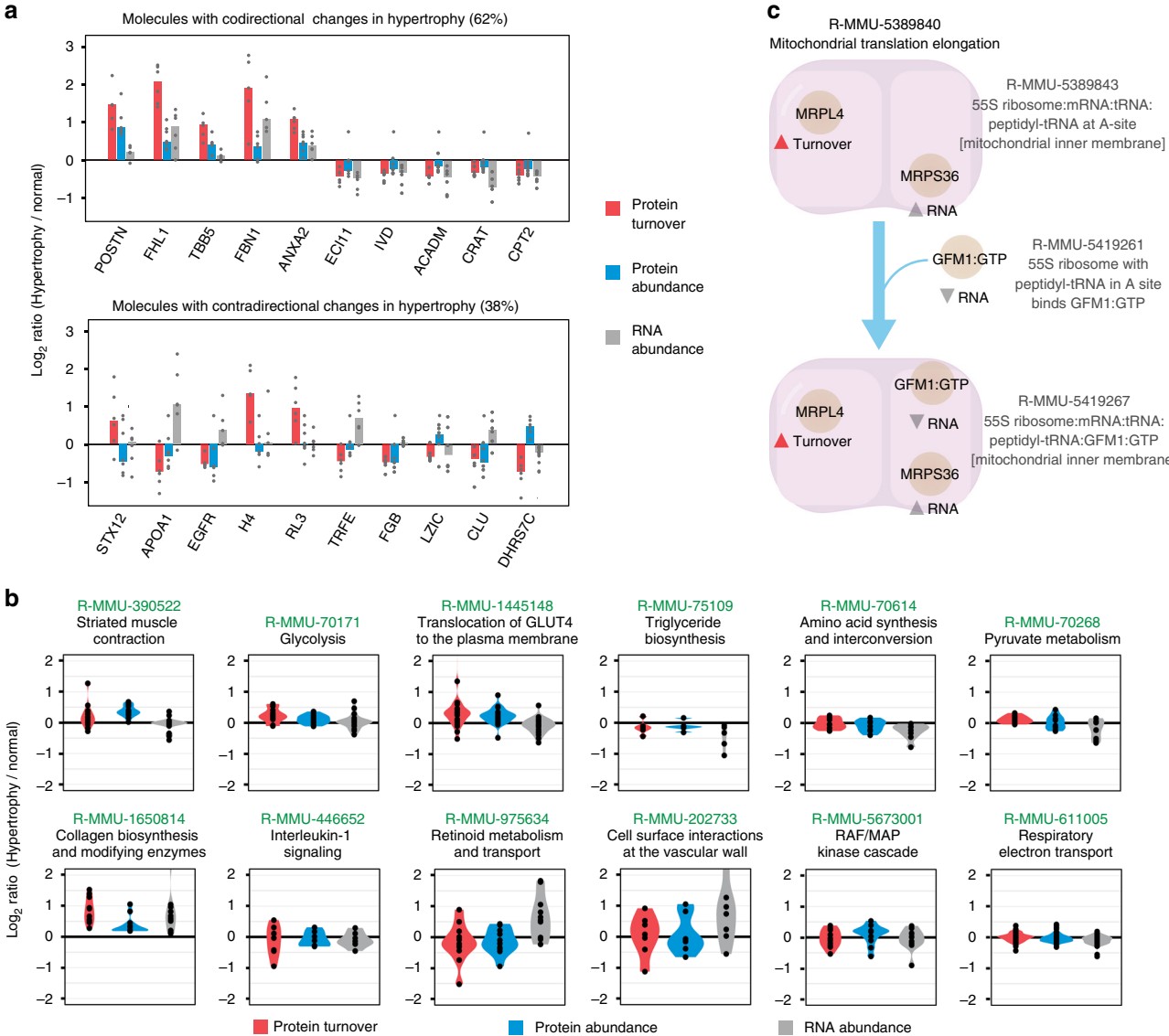

**Fig. 2** Differential regulation of genes and pathways in cardiac hypertrophy. **a** Bar charts of log ratio of changes of selected genes (*y*-axis) in protein turnover (red), protein abundance (blue), and transcript abundance (gray) levels (*x*-axis). Both co-directional changes (left panel) and contra-directional changes in the three measured parameters are found. FHL1, POSTN, TBB5, FBN1, ANXA2 are increased in all three parameters; ECI11 and others decreased in all three parameters. Proteins, including STX12 and APOA1 show contra-directional changes, suggesting the non-overlap of implicated disease genes is not due to arbitrary cutoffs. **b** Distribution of log ratios (hypertrophy/normal) in protein turnover (red), protein abundance (blue), and RNA abundance (gray) for protein members (data points) in 12 Reactome pathways relevant to cardiac functions. **c** Example Reactome pathway (R-MMU-5389840 Mitochondrial translation elongation) implicated in hypertrophy from the multi-omics data. MRPS36 and GFM1 show reproducibly altered transcript abundance, whereas MRPL4 is implicated via turnover

among which 29 also show differential expression in their steady-state transcript or protein abundance (Supplementary Data 1). The remaining 70 proteins implicated via turnover alone (vs. 273 proteins) therefore represent a 34.5% gain in disease signature discovery over steady-state protein/transcript measurements alone, and a 75% gain in discovery if only protein abundance were considered (vs. 116 proteins). Remarkably, only three signatures show consistent increase of transcript abundance, protein abundance, and protein turnover: annexin II (ANXA2), annexin III (ANXA3), four-and-a-half LIM domains protein 1 (FHL1). Annexins are calcium-dependent phospholipid-binding proteins in cytoskeleton and membrane remodeling that are known to be up-regulated in heart failure at the transcript level[30]. FHL1 is a sarcomeric biomechanical stress sensor that is implicated in cardiac hypertrophy[31]. The multi-omics here

indicates they are reliable hypertrophy markers on both transcript and protein levels, whose induced transcript expression is matched by increased protein synthesis and sustained increase of protein abundance. Conversely, a considerable proportion (38%) of disease signatures show inconsistent or discordant directions in their changes of transcript abundance, protein abundance, or protein turnover in hypertrophy (Fig. 2a). For instance, dehydrogenase/reductase SDR family member 7C (DHRS7C) shows consistently repressed turnover amid isopro-terenol challenge in five out of six mouse strains (1.3–2.7-fold) whilst its protein abundance is increased (1.2–1.7-fold). DHRS7C is a cardiomyocyte-expressed ER dehydrogenase previously reported to be transcriptionally downregulated in heart failure[32]; we speculate that this transcriptional downregulation might be balanced by decreased protein degradation, leading to protein

accumulation. Indeed, we found that under treatment of epoxomicin, a specific proteasome inhibitor, 83% of measured proteins showed decreased turnover rates, confirming the measured dynamics is influenced in part by protein degradation mechanisms (Supplementary Fig. 4). Taken together, contra-directional molecular profiles indicate that the non-overlap of signatures is unlikely to be due to arbitrary magnitude cutoffs chosen in the analysis.

Cellular pathways relevant to cardiac functions show diverse changes across omics data type (Fig. 2b), whereas signatures nominated from each omics type are enriched in different functional categories. Turnover signatures are enriched with proteins located in the extracellular matrix (Enrichment ($E$): +6.6-fold; hypergeometric test with Benjamini–Hochberg adjusted $P$ value ($P$): 3.2e-3), sarcolemma ($E$: 7.5; $P$: 2.8e-3), cellular potassium ion homeostasis ($E$: 39.5; $P$: 2.4e-3), glycolysis ($E$: 8.3; $P$: 2.2e-2). Proteins with consistently decreased turnover are more likely to localize to the mitochondrial matrix ($E$: 4.9; $P$: 1.7e-3) and function in fatty acid beta-oxidation ($E$: 10.9; $P$: 3.9e-3). Signatures with increased protein abundance in hypertrophy are statistically more likely to function in muscle contraction ($E$: 16.2; $P$: 7.1e-5), whereas those with decreased abundance are enriched in peroxisomal membrane ($E$: 8.3; $P$: 8.7e-3). Transcript up-regulation is observed preferentially in extracellular space ($E$: 9.2; $P$: 6.1e-12), cell surface ($E$: 4.6; $P$: 3.1e-3), regulation of inflammatory response ($E$: 19.5; $P$: 1.5e-3), and proteolysis ($E$: 4.5, $P$: 8.6e-3). Downregulated transcripts are preferentially mitochondrial proteins ($E$: 3.4; $P$: 2.8e-4), involved in fatty acid beta oxidation ($E$: 9.4; $P$: 1.0e-2). Hence, the data here suggest that disease signatures associated with different aspects of cardiac functions may be discovered from each omics data type, and are consistent with specific regulatory modalities in each cardiac subproteome.

At the same time, the integration of multi-omics data facilitated both disease signature discovery and interpretation via pathway annotations. Further analysis on the 70 proteins discovered via turnover alone (37 proteins with increased turnover and 33 with decreased turnover) suggests they also contain proteins that are functionally associated with the remaining 203 candidate disease proteins (out of 273). (Supplementary Fig. 5). The inclusion of turnover data, therefore, led to a larger-than-expected impact on the number of annotated cellular pathways that may be associated with the disease signatures. A total of 36 nonredundant pathway groups with 3 or more disease protein members are implicated in the multi-omics experiment, representing a 57% gain over the 23 pathways discovered absent turnover data (Fig. 1a; Supplementary Data 2). For instance, the Reactome pathway R-MMU-5389840 (Mitochondrial translation elongation) is implicated via the disease proteins MRPS36, GFM1, and MRPL4. MRPL4 was selected as a disease marker due to its consistently increased turnover upon isoproterenol stimulation across all six genetic strains, offering evidence for its involvement in disease (Fig. 2c). Therefore, turnover data complement transcript and protein abundance data to nominate the mitochondrial translation elongation pathway as being involved in cardiac hypertrophy. In another example, data from ENO1 and PFKP (both with increased turnover) as well as PGAM1 (increased transcript and turnover) jointly implicate the glycolysis pathway (R-MMU-70171) in hypertrophy. Consistent with functional association, we identified 54 physical protein–protein interaction pairs that occur between the 70 turnover disease proteins and the remaining 203 candidate disease proteins (Supplementary Data 3), e.g., ubiquitin ligase HUWE1 (decreased expression in hypertrophy) with ubiquitin hydrolase USP9X (decreased turnover). These observations are preserved in other pathway definition cutoffs (e.g., 9 implicated pathways with

turnover data vs. 5 without, when a nominated pathway requires ≥ 5 disease proteins), altogether suggesting that the disease burden of various cellular pathways may be under-estimated in current discovery experiments.

**Meta-analysis with existing transcriptomics data**. The distinct-ness of the omics data types is corroborated when the multi-omics data are analyzed against existing transcriptomics data sets in the public domain[28,33–47]. To determine how the candidate disease signatures above are relevant to previously nominated cardiac disease genes[28,33–47], we performed a meta-analysis on multiple data sets representing surgical or pharmacological models of pathological cardiac hypertrophy (Fig. 3a). Using the keywords "heart" and "hypertrophy", we queried NCBI Gene Expression Omnibus (GEO) and retrieved 55 qualifying data sets from 21 GEO data series, from which we compared the data in 29 data sets that passed data quality and experimental design filters (see Methods) (Fig. 3b). The gene lists were subsequently aggre-gated by considering their overlaps in top-ranking genes against permutations[48], in order to evaluate the rank similarities in changes in transcript abundance, protein abundance, and protein turnover. We found consistent similarities among transcriptomics data sets as determined by robust correlation (correlation coef-ficients: −0.1–0.3), whereas protein-level changes were only modestly correlated with transcripts: only 18% of compared data sets showed significant rank similarity in the order of the tran-script/protein induction/repression (Fig. 3d). In other words, alterations in protein turnover were not predictable from the retrieved transcriptomics data sets. Among the 99 proteins with consistently altered turnover in hypertrophy, ANXA2, ANXA3, and FHL1 show uniform changes in protein turnover, abundance, and transcripts across all data sets, whereas 22 proteins including CLU, ENO1, DHRS7C, PLG, and MRPL4 show discordant reg-ulations between turnover and transcript abundance during hypertrophy (Fig. 3e) (Supplementary Data 4). These results affirm that protein-level data reveal independent insights into pathogenic processes beyond existing transcriptomics studies.

**Co-regulated turnover of functionally associated proteins**. To gain insight into how protein turnover data may reveal additional disease signatures, we examined the turnover characteristics of different cellular components. A co-clustering of all three omics parameters segregated cardiac proteins into distinct clusters of behaviors, which may be classified into one of three archetypes (Fig. 4a). In Type I clusters, steady-state protein levels correspond largely to transcript abundance. Type II proteins are characterized by relatively low transcript levels and protein abundance, hence, may be regulated post-transcriptionally or post-translationally via protein synthesis and degradation rates. Type III proteins have high transcript and protein abundance with relatively low turn-over rate constants, containing many housekeeping proteins including basal metabolic proteins that showed low production rates. We observed that Type I and Type II proteins differed from Type III proteins in that they exhibited greater expression changes during hypertrophy (Fig. 4b). Multivariate linear regression analysis suggests that the turnover rate of a protein predicts the extent of its relative expression change at day 1 as well as day 14 of isoproterenol stimulation (Fig. 4c). This rela-tionship is independent from protein baseline abundance, which is by itself an indicator of the magnitude of differential expression such that low-abundance proteins appear more responsive to hypertrophic stimuli. Unlike turnover, baseline abundance does not predict how quickly the protein alters in early hypertrophy (on day 1 of isoproterenol). Taken together, the data suggest a proteome architecture reminiscent of previously hypothesized

signaling functions of protein turnover, in which the rate at which proteins can equilibrate to new steady states is primarily a function of the half-life of the protein pool[22]. Indeed, when we examined the biological functions of proteins with different turnover rates, we found that metabolic and housekeeping proteins including respiratory complexes and tricarboxylic acid cycle

proteins are the most long-lived, and overrepresented among proteins with turnover rates in the range of 11 days, or longer. Slow-turnover proteins are also highly expressed, consistent with the notion of low protein flux due to a plateau of production rates[7]. The next most stable groups contain ribosome translational and glycolytic proteins, followed by proteins in proteolysis,

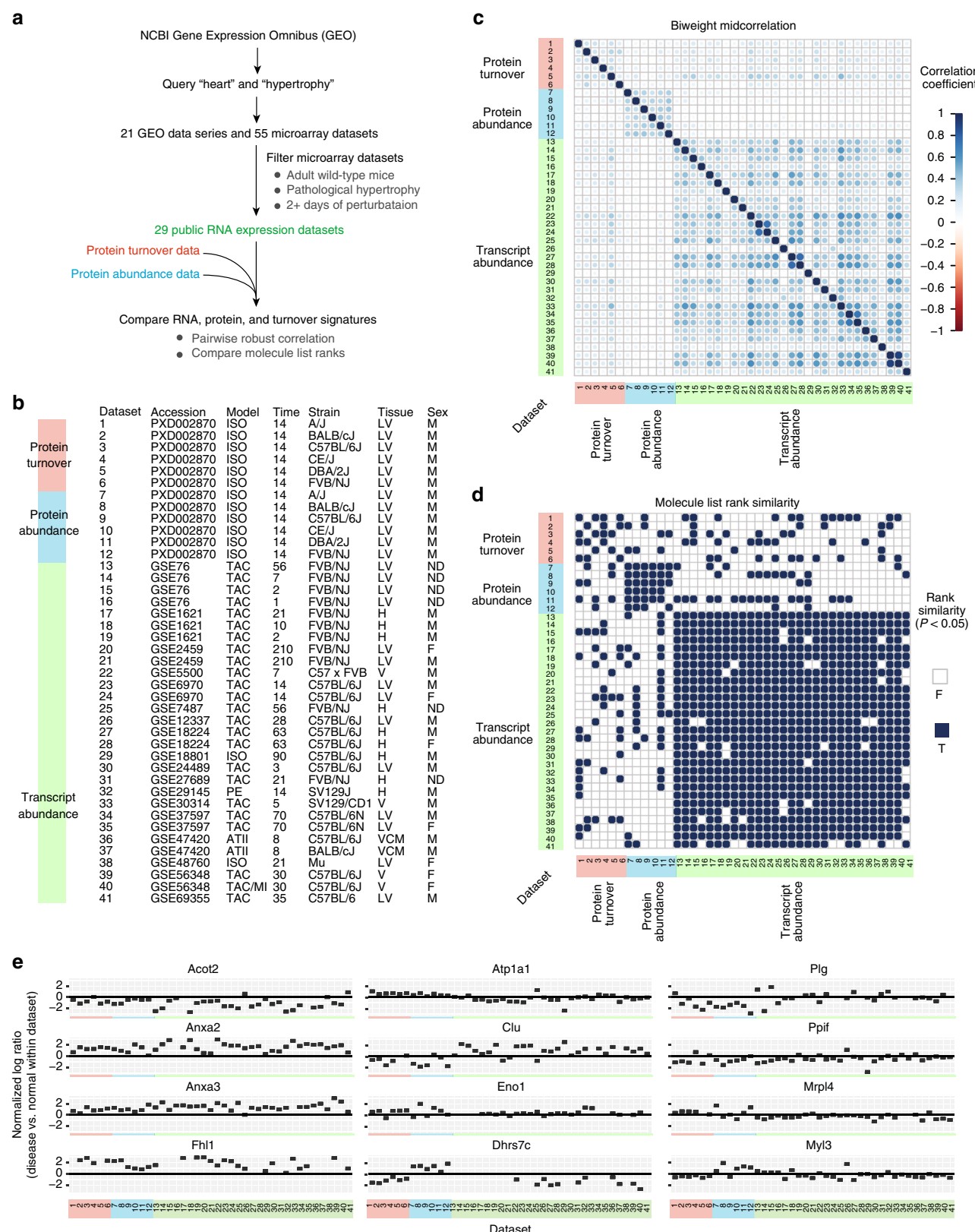

ubiquitin-mediated catabolic processes, protein folding, or transport. In contrast, fast-turnover proteins are overrepresented in signal transduction, small GTPase-mediated signaling, vesicle-mediated transport, translational initiation, and nucleocytoplasmic transport processes (Fig. 3d). The examined cluster classifications dissipate upon removal of turnover data, suggesting turnover provides essential information in parallel to transcript and expression data in the examined system.

To further examine the contribution of protein turnover data to disease signature discovery, we asked whether variations in protein dynamics may be associated with the variable cardiac phenotypes among the mouse strains. In the analyses above we considered the largely consistent molecular changes across mouse strains. It was evident however that subtle inter-strain variations in protein turnover also exist. Although all strains are considered healthy wild-types, they carry heritable genetic variations in cardiac traits and functions in response to isoproterenol stimulation[27], e.g., A/J mice showed greater elevation in Na$^+$/K$^+$ ATPase turnover upon isoproterenol compared to other strains (P: 0.022) (Supplementary Fig. 2). Such inter-strain differences in turnover permits linear regression models to be applied to identify potential associations between protein turnover and cardiac phenotypes. We organized the data into a robust correlation matrix of the turnover of each protein across strains and disease status, then clustered the proteins by weighted gene correlation network analysis[49,50]. Proteins are thereby classified into modules in an unsupervised manner (Fig. 5a), although no inference is made on network topology due to sample size. Regression modeling against normalized heart-weight-over-body-weight ratios (HW/BW) (Fig. 5b) revealed that the majority of cardiac proteins cluster into a module (turquoise) of 527 proteins whose turnover rates are positively correlated with hypertrophy across the mouse strains (Fig. 5c). The turquoise-module proteins are enriched in biological processes including cell adhesion (GO:0007155; E: 3.7; Benjamini–Hochberg adjusted P value (P): 2.9e-5), glycolytic process (GO:0006096; E: 11.0; P: 3.8e-4), actin filament organization (GO:0007015; E: 5.0; P: 2.6e-3), translation (GO:0006412; E: 2.0; P: 4.5e-3), and sodium ion transport (GO:0006814; E: inf; P: 1.1e-2), altogether suggesting the hypertrophic phenotype, i.e., increased HW/BW, is associated with faster turnover of extracellular matrix remodeling, contractility proteins, and translational regulators. By contrast, a module of 129 proteins is negatively correlated with the HW/BW at the examined time points during isoproterenol challenge (Fig. 5d). Proteins in this module are enriched in the mitochondrial matrix and in fatty acid metabolic process (GO:0006631; E: 5.9; P: 7.3e-5) as well as oxidation-reduction process (GO:0055114; E: 3.0; P: 9.0e-5). Among the proteins in the module are a striking complement of members in the mitochondrial fatty acid oxidation pathway including ACAT1, ACAD10, HADHA,

HADHB, DBT, ACAA2, ETFA, and ACSL1. Several of the blue-module proteins show consistent alterations in at least four strains in the candidate disease protein analysis above (e.g., ACADM and ACOT2), but other proteins co-vary with HW/BW in the strains without consistent changes in 4+ strains (e.g., ACAA2 and ACSL1). The observed protein–trait relationships were preserved in the log-turnover space (Supplementary Fig. 6) and also when we cross-referenced the turnover networks with a prior data set on echocardiography measurements from an independent cohort of mice in identical strains upon isoproterenol challenge[27] (Supplementary Fig. 7a-b), suggesting the HW/BW data and protein–trait relationship are robust. The results are robust to network construction parameters, and are reproduced when additional data from a smaller data set previously acquired on outbred Hsd:ICR mice were appended (Supplementary Fig. 7c-d). Unlike turnover, the steady-state abundances of protein modules are not significantly associated with HW/BW. Hypertrophy is known to promote glycolysis and suppress fatty acid oxidation in the heart[51,52], but expression profiles are unclear on this shift. Glycolytic proteins are generally not found to change in abundance in hypertrophy[52,53]. Clustering analysis with protein abundance revealed few clear relationships with heart traits (Supplementary Fig. 7e-f). Taken together, these data implicate novel associations between metabolic remodeling in hypertrophy and coordinated protein turnover in selected subproteomes.

Lastly, we observed co-regulated turnover between functionally related proteins. Protein–protein interacting partners are known to be co-regulated in expression and share similar abundance, such that gene coexpression across species or cell states are sometimes exploited to predict physical or functional associations[54]. To determine whether interacting partners also share similar turnover rates, we retrieved 10,279 known protein–protein interactions from public interactomes (IntAct, BioGrid, MintDB), and compared the turnover differences of interacting pairs to those of simulated random pairs or permuted interactomes. In every queried database, bona fide interacting partners had more similar turnover than random and permuted pairs (Mann-Whitney test P < 2.2e-16, Komogorov-Smirnov P < 2.2e-16, permutation test P < 1e-5) (Fig. 6a, b). Known protein–protein interacting pairs show similar turnover, including PKA and its anchoring protein AKAP1; and protein phosphatase 2A and its regulatory subunit (Fig. 6c; Supplementary Data 5). On average, protein–protein interaction pairs differ by 1.62-fold (interquartile range IQR: 1.23–2.68) in turnover, vs. 1.92-fold in permuted interactomes (IQR: 1.35–3.23) and 2.21-fold in random pairs (IQR: 1.44–3.97), suggesting a propensity for similar turnover between interacting partners (Fig. 6d). Among analyzed interaction pairs, 64% were between proteins with <2-fold differences in turnover, vs. 44% in random pairs

**Fig. 3** Protein turnover data are not predictable from accessible transcriptomics data sets on cardiac hypertrophy and heart failure. **a** Schematic of integrated analysis of the protein turnover and protein abundance data from the six mouse strains alongside 29 sets of the publicly accessible transcriptomics data on NCBI Gene Expression Omnibus (GEO). **b** Summary of retrieved data sets. Accession: accession number of the NCBI GEO data set. Model: experimental model to induce cardiac hypertrophy—TAC, transverse aortic constriction; MI, myocardial infarct; ATII, angiotensin II stimulation; ISO, isoproterenol stimulation. Time: days of experimental model at sample collection. Species: Mm, *Mus musculus*. Strain: Genetic background of the animals studied—Mu, multiple strains. Tissue: cardiac tissues collected for profiling—LV, left ventricle; H, whole heart; V, whole ventricle. Sex: gender of animals—M, male; F, female; ND, not determined/reported. **c** Each cell in the matrix is a pairwise comparison between two data sets. Color denotes the strength of robust biweight midcorrelation between the resulting signature lists (ratio of disease/control) between the compared data sets. Transcript abundance data sets (green labels) show appreciable correlation with one another but not protein abundance (blue labels) and turnover (red labels) data. **d** Comparison of gene list rank similarity metrics[48] of the retrieved protein abundance, turnover data, and accessible transcriptomics data sets. Solid squares: significant protein list order similarities between two data sets (P < 0.05). **e** Normalized log ratio (*y*-axis) of selected proteins across data sets. Hypertrophy markers ANXA2, FHL1 and ANXA3 show consistent differential regulations in all experiments, whereas 22 of 95 examined proteins are implicated in hypertrophy via turnover but have no discernible, or discordant, changes in transcript abundance. Hence over existing transcript data sets, protein turnover measurements provide independent information on cardiac hypertrophy

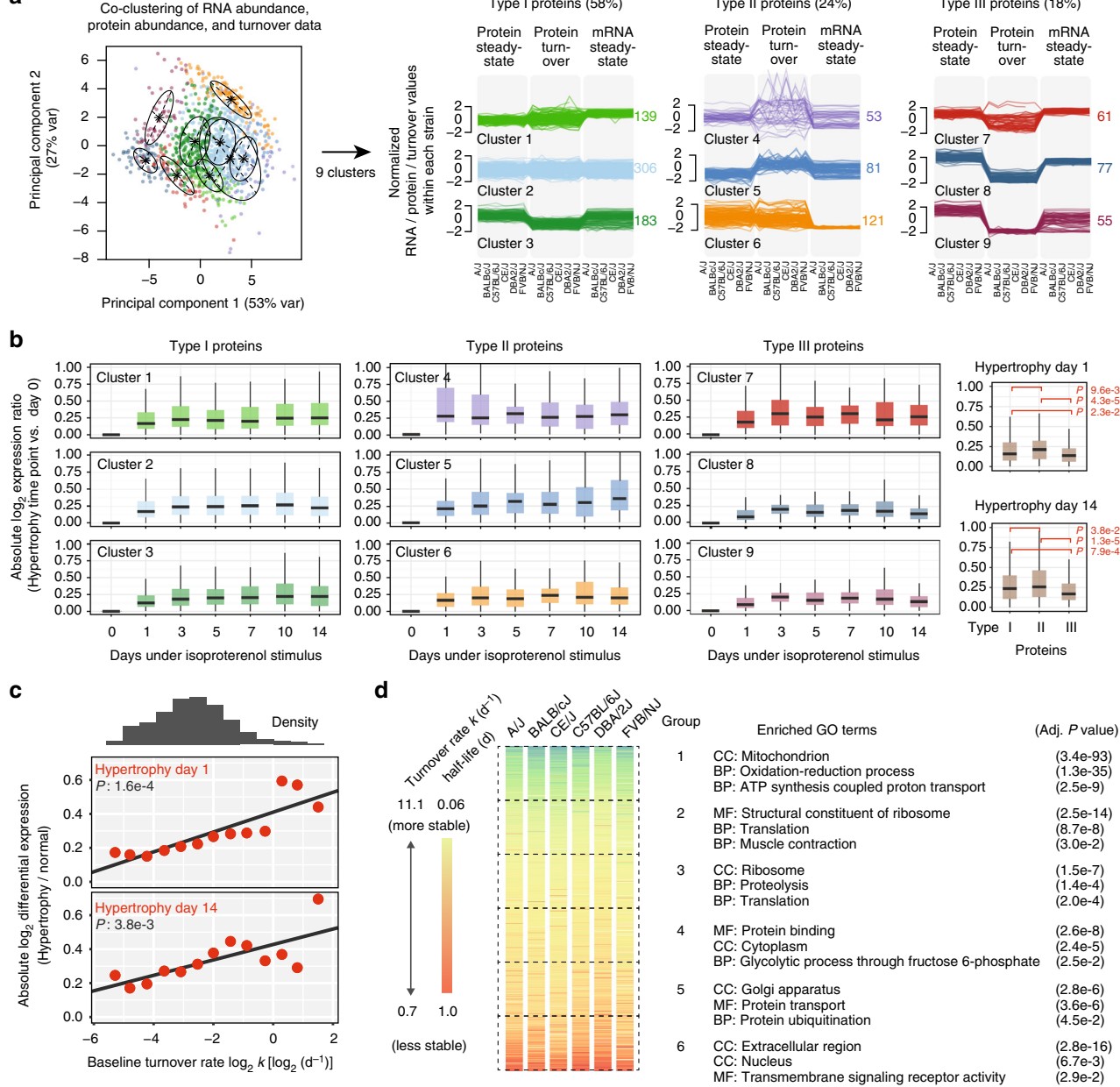

**Fig. 4** Co-clustering of RNA, protein, and the abundance data reveals hierarchical protein response to cardiac hypertrophy. **a** Unsupervised clustering of the multi-omics data sets distinguishes genes into 9 clusters based on all three omics data types. First two principal components (PC) are shown; Ellipses demarcate the locations of clusters. Right: The 9 identified clusters of proteins are visualized separately with line charts showing standardized protein abundance, protein turnover, and transcript abundance in all strains. (Left) Type I protein clusters have relative protein abundance commensurate with relative protein turnover and transcript abundance. Type II proteins show relatively high turnover. Type III proteins have relatively high transcript and protein abundance but low turnover. Numbers: protein counts in clusters. **b** Boxplots of log ratios of protein abundance in hypertrophic vs. normal hearts (*y*-axis) at multiple time points (*x*-axis) from each cluster, showing their different responses to isoproterenol stimulation over 14 days. Type II proteins exhibit earlier and greater responses in protein abundance changes during hypertrophy compared to Type I proteins, whereas Type III proteins show suppressed response. The data suggest an organization pattern of protein turnover associated with the speed and magnitude of protein remodeling in the heart. Inset shows side-by-side comparisons of Type I, II, III proteins. Box: 25th to 75th percentile; whiskers: 5th to 95th percentile. **c** Higher baseline turnover rate (*x*-axis) of a protein is associated with greater magnitude of differential expression under hypertrophy (*y*-axis). *P* value: linear regression of absolute differential expression against baseline turnover rates in all proteins, with baseline relative abundance of a protein as covariate. **d** Heat map showing distributions of turnover rates of cardiac proteins quantified across mouse strains (green: slower turnover/longer half-life; red: faster turnover/ shorter half-life). Proteins are divided into six bins (Group 1–6) according to their average turnover. Each bin is significantly enriched in distinct Gene Ontology terms that reflect the organizational structure of the proteome. Housekeeping proteins tend to turn over slowly and regulatory proteins turn over quickly

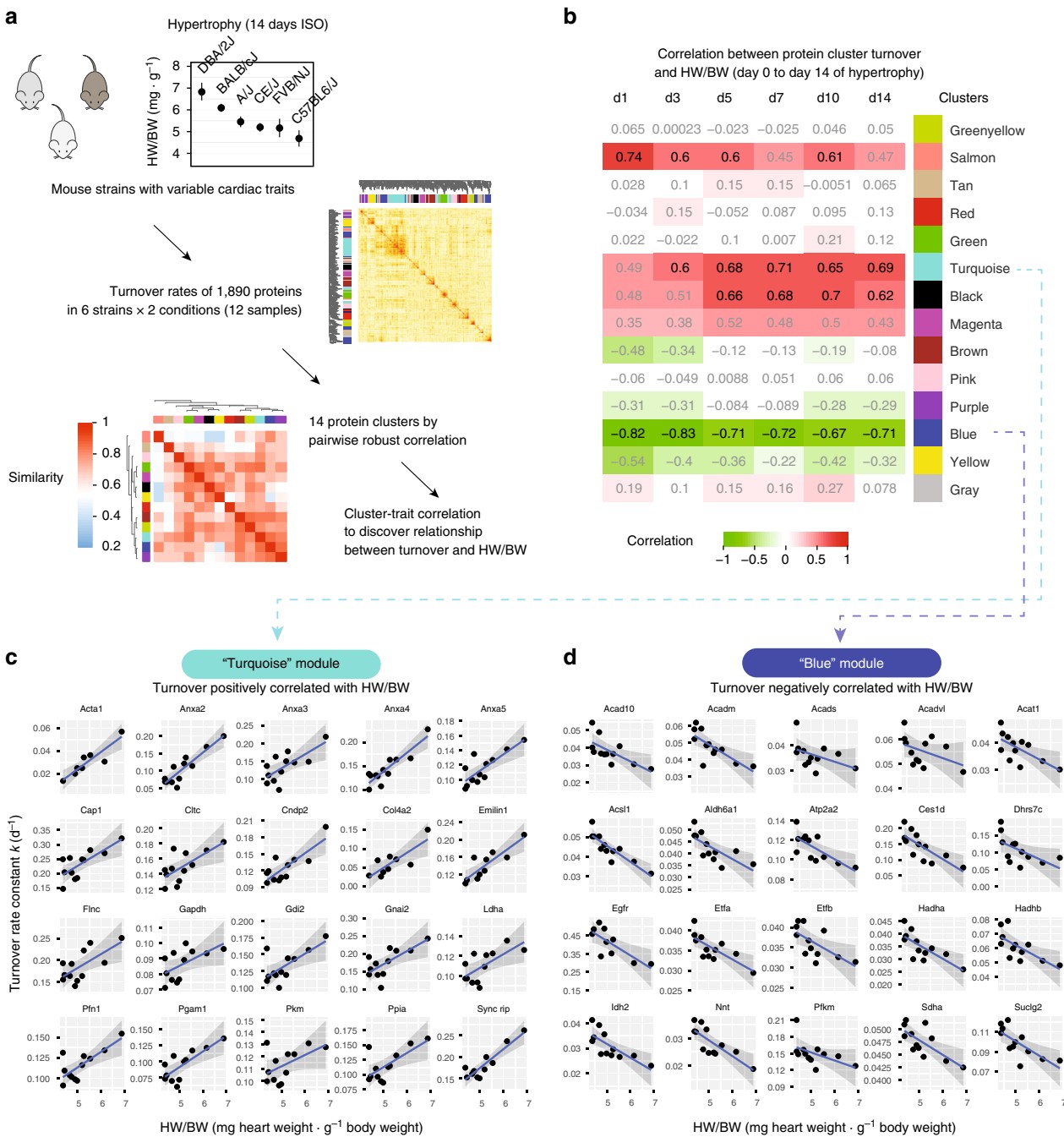

**Fig. 5** Coordinated temporal dynamics of protein clusters in cardiac hypertrophy. **a** The six mouse strains show differential heart-weight-to-body-weight ratios (HW/BW; y-axis; error bars: s.d.) over 14 days of beta-adrenergic stimulation (x-axis). Turnover rates of 1890 proteins in the six genetic strains in normal and hypertrophy are clustered based on their normalized turnover rates into 14 modules. **b** Correlation between cluster protein turnover and HW/BW among the six genetic strains during hypertrophy. Heat map colors and numbers denote correlation coefficient ρ. Black texts: suggestive correlation ($P < 0.05$; Benjamini–Hochberg adjusted $P > 0.05$); Bold texts: significant correlation (adjusted $P < 0.05$). **c** Turnover of proteins in the module labeled "Turquoise" is positively correlated with hypertrophy. The module includes membrane remodeling proteins (ANXA2, ANXA5), extracellular proteins (e.g., COL4A2), and glycolytic proteins (e.g., LDHA, PGAM1) (20 representative proteins shown). The data points: $\log_2$ turnover rates (y-axis) vs. HW/BW (mg per g) (x-axis). Blue line: local regression; area: confidence interval. **d** Turnover of proteins in the "Blue" module is negatively correlated with hypertrophy. The module is highly enriched in fatty acid oxidation proteins including ACAD10, ACAT1, ACSL1, ETFA, ETFB, and HADHB

(hypergeometric test $P < 2.2e$-16). This observation is preserved after correcting for protein abundance and protein localization (Fig. 6e). Moreover, the synchronizing relationship between protein–protein interaction and protein turnover became stronger during hypertrophy ($P$: 1.4e-9) (median 1.58-fold differences in real pair vs. 1.90-fold in permuted pairs) (Fig. 6f). To further examine this relationship, we retrieved 357,138 predicted protein associations from an independent data set on STRING database[55]. STRING integrates multiple data sources and summarizes the confidence of predicted protein associations with a score ranging from 150 to 999. We found that turnover rate difference is significantly associated with the confidence score of protein association on STRING (Fig. 6g). Taken together, the data suggest that functionally associated proteins are co-regulated in turnover.

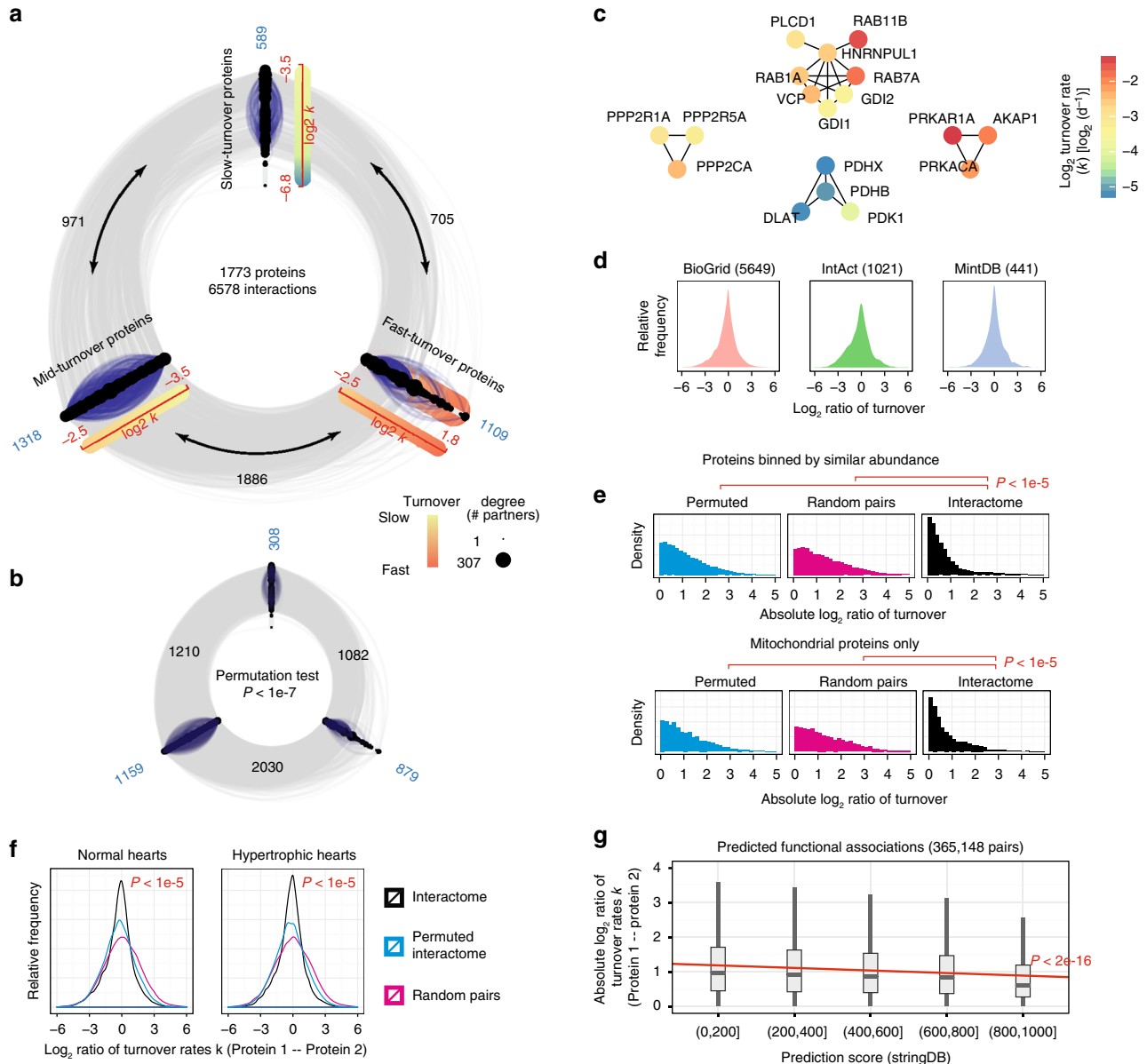

**Fig. 6** Protein–protein interacting partners are coordinated in turnover. **a** Network diagram of 6578 annotated protein–protein interactions involving 1773 anotated proteins with turnover rates measured in ≥4 mouse strains, retrieved from three databases (BioGrid, IntAct, and MintDB). Protein nodes in the network are grouped by turnover rates and aligned along three axes (slow, mid, and fast). Edges: interactions between two proteins in the same (blue) or different (gray) turnover rate groups. Numbers denote the number of interaction edges within each group (blue) and between groups (black). Radius of protein nodes: degree of interacting partners. **b** Permutation means of inter-group and intra-group connections, showing that interactions preferentially occur among proteins with similar turnover rates ($P < 1e-7$ from 1e7 permutations). **c** Selected examples of interacting proteins with similar turnover. **d** Interacting partners have significantly more similar turnover, from the data individually curated in each of the 3 queried databases. **e** Distribution of $\log_2$ protein turnover differences in interaction partners, random pairs, and permuted pairs, controlling for the effect of protein abundance by including only a bin of 250 proteins with moderate abundance (relative abundance $2^{-11.9}$–$2^{-10.7}$). The 250 proteins formed 2043 interactions, with 1.26-fold average turnover differences vs. 1.88-fold in random pairs (permutation test $P < 1e-5$). Interacting partners likewise have similar turnover when only mitochondrial proteins are analyzed to control for the effect of subcellular localization (permutation test $P < 1e-5$). **f** Distribution of $\log_2$ turnover ratios between retrieved interacting partners (black), random pairs of analyzed proteins (magenta), and randomly resampled interaction pairs (blue) (permutation test $P < 1e-5$) in normal and hypertrophic hearts. **g** Differences in protein turnover rates (y-axis) are correlated with the predicted interaction score between two proteins on STRING (x-axis), with every 1-fold difference in turnover reducing the score by 22 (linear regression $P < 2.2e-16$). High-confidence predicted associations (scores of 800–1000) are particularly more similar in turnover than protein pairs in other bins (Kruskal–Wallis test with post hoc $P < 2.2e-16$). Box: 25th to 75th percentile; whiskers: 5th to 95th percentile

## Discussion

Here we integrated experimental and data science approaches to analyze new and existing data sets on protein turnover, protein abundance, and transcript abundance in normal and hypertrophic hearts. Each molecular feature revealed partial aspects of disease development, from all of which combined we assembled a portrait of cardiac pathogenesis, in which a global increase in protein-level abundance of both contractile and sarcolemmal apparatus is counterposed by relative reduction in mitochondrial metabolic machinery at the transcript and protein turnover levels.

Our results lend credence to the notion that both transcriptional and post-transcriptional regulations influence complex disease development and may be integrated in biomedical inquiries.

In particular, we identified 70 candidate disease signatures by turnover measurements which show no overt changes in their steady-state transcript and protein abundance. Meta-analysis of accessible RNA-seq data sets from independent cohorts suggests that the data variations cannot be fully explained by experimental noise; the turnover candidate disease signatures were not predictable from existing transcriptomics data sets. Overall there was poor correlation in the variations of protein response to isoproterenol stimulus at the transcript, protein, and turnover levels. Recent data also affirm that candidate disease gene signatures for cardiac hypertrophy and failure had limited commonality at the transcriptome and the proteome levels, even when intra-sample variabilities are accounted for[56]. Indeed, we found that measurements of post-translational states of gene expression can expand the pool of candidate disease signatures in practice and enable more comprehensive understanding of stimulus responses than attainable from transcript data alone[7]. From the analysis of multiple mouse strains, the turnover data further indicate that the dynamics of glycolytic and fatty acid oxidation pathways may be a contributing factor to cardiac substrate utilization phenotypes. We observed coordinated turnover regulation at multiple levels of proteome organization. Protein–protein interacting partners synchronize in turnover in the normal and hypertrophic heart, whereas proteome remodeling in hypertrophy is associated with simultaneous perturbations of multiple members in annotated pathways.

The observation of distinct signatures from transcript abundance, protein abundance, and protein turnover is consistent with known mechanisms of post-transcriptional and post-translational protein regulations. Although the central dogma of molecular biology posits that genetic information ultimately translates into information in amino acid sequences, various steps between genes, mRNA, and proteins assert regulatory contributions that can lead to imperfect correlation between genetic programs and protein molecular phenotypes[8], including post-transcriptional and post-translational controls[7,9]. Whether transcriptional controls predominate during dynamic response to stimuli has been observed to differ by experimental model—whereas response to lipopolysaccharide stimulus in dendritic cells may be induced largely by transcriptional processes[11], translational processes play crucial roles in cell-wide proteostatic scenarios such as endoplasmic reticulum stress in HeLa cells[10]. Changes in protein abundance may be uncoupled from changes in protein turnover by mechanisms of protein degradation, which are evidently fine-tuned by the fact that the human genome encodes 600–1000 E3 ubiquitin ligases. When protein degradation is stimulated, a measurable increase in turnover may result in parallel with a decrease in overall protein abundance[21]. In the heart, mounting evidence on the coordinated turnover of myofibrillar proteins, metabolic enzymes, and transcription factors by the muscle-specific TRIM ubiquitin E3 ligases MuRF1/2/3 also suggests cardiac protein turnover regulators can target selected sub-proteomes and exert post-transcriptional regulatory effects on protein expression[57–60].

Large-scale proteome turnover studies have advanced our understanding of health and diseases, and now allow post-transcriptional regulations of individual genes to be detected in a large scale in physiologically relevant in vivo models. Protein turnover measurements have been applied to animal models of cardiac hypertrophy, aging, and caloric restriction, providing unique information on gene regulation in these biological processes. In this study, we assessed protein turnover by modeling percentage label incorporation as a proportion to the total protein pool over time. When the heart protein pool is expanding in size such as during gene up-regulation, physiological growth in animal mass, or pathological growth (hypertrophy), one might assume net protein synthesis rate outpaces degradation rates. The measured parameter therefore reflects the degenerate combinations of net absolute synthesis and degradation rates. Even under non-constant protein pool sizes, we show that two-compartment kinetics effectively models isotope incorporation with no loss in precision, supporting confident relative comparisons between normal and diseased heart for disease signature discovery. Future experiments might combine stable isotope labeling with absolute protein quantification to measure absolute synthesis and degradation rates, which can provide further information on the mechanism of post-transcriptional regulation. Nevertheless, the data presented here already demonstrate the potential to exploit post-transcriptional dynamics to identify disease signatures, and pave the way for further studies to fine-map the mechanisms of gene regulation such as in other disease models and individual cell type studies.

Taken together, our results provide a framework for deeper interrogation into the temporal dynamics of proteome remodeling in a complex disease model. We suggest that multi-omics investigations of proteome remodeling dynamics in complex systems may yield new research avenues into pathogenic mechanisms.

## Methods

**Summary**. Protein turnover was traced by the introduction of deuterium atoms to the drinking water of animals, which equilibrated with amino acids via biosynthesis and deamidation reactions and subsequently became incorporated into proteins upon continuous protein synthesis and degradation. The degree of incorporation was measured by high-resolution mass spectrometry. We use a software platform we previously developed to automate the interpretation of deuterium-labeled mass spectra and analyze deuterium incorporation, followed by kinetic modeling to measure protein stability as the proportion of proteins replaced per day, i.e., turnover rate.

**Reagents and chemicals**. $^2H_2O$ (70% and 99.9% molar ratio) was from Cambridge Isotope Laboratories and filtered through 0.1-μm polyethersulfone membranes (VWR International). Milli-Q (Millipore) filtered water (18.2 MΩ) was used. Chemical reagents were from Sigma-Aldrich unless specified.

**Experimental animal models**. Animal experiments were performed in accordance with the Guide of the Care and Use of Laboratory Animals by the National Research Council and approved by UCLA Institutional Animal Care and Use Committee. C57BL/6J, CE/J, A/J, DBA/2J, FVB/NJ and BALB/cJ mice (male, 9–12 weeks of age) were purchased from Jackson Laboratories. Upon arrival, the animals were housed in a 12 h/12 h light-dark cycle (lights on from 6 a.m. to 6 p.m. local time) with controlled temperature, humidity, and free access to standard chow and water. Following acclimatization, the animals received two intraperitoneal injections of 500 μL 99.9% (molar ratio) $^2H_2O$-saline 4 h apart to initiate labeling. Mice were then given free access to 8% (v/v; 7.25% molar ratio) $^2H_2O$ in the drinking water supply. Independent groups of 3 mice each were euthanized at 0, 1, 3, 5, 7, 10, 14 days following the first $^2H_2O$ injection of each group at 12:00 noon for sample collection. Labeling began independently for each group. Sample size (7 time points × 6 strains) were chosen based on previously measured variance in measured turnover rates in mice[25]. The time points were chosen based on previous experience of average protein turnover in the mouse heart (median half-life of 7 days). Animals in the hypertrophy group were randomized by the surgeon to initiate cardiac hypertrophy via surgical implantation, concurrently to labeling initiation, subcutaneous micro-osmotic pumps (Alzet) calibrated to deliver 15 mg kg$^{-1}$·d$^{-1}$ isoproterenol over 14 days[61]. Measurements of heart-weight-to-body-weight ratios (HW/BW) were recorded by a blinded observer.

**Protein sample extraction and digestion**. Mouse hearts were excised and homogenized with a Dounce homogenizer (20 strokes, 4 °C) in an extraction buffer: 250 mM sucrose, 10 mM HEPES, 10 mM Tris, 1 mM EGTA, 10 mM dithiothreitol, protease and phosphatase inhibitors (Pierce Halt), pH 7.4. The homogenate was centrifuged (800 $g$, 4 °C, 7 min). The pellet was collected as the nuclear and extracellular fraction. The supernatant was centrifuged (4000 $g$, 4 °C, 30 min) and collected as the organelle-depleted intracellular fraction. The pellet was washed and centrifuged again to collect as the mitochondrial and microsomal fraction. Protein concentrations were measured by bicinchoninic acid assays

(Thermo Pierce). Extracted protein fractions were solubilized with RIPA then digested on-filter using 10,000 Da polyethersulfone filters (Nanosep; Pall Life Sciences). Sample buffer was exchanged on-filter with ammonium bicarbonate (100 mM, 100 μL). The samples were reduced (70 °C, 5 min) with dithiothreitol (3 mM) and alkylated in the dark (ambient temperature, 30 min) with iodoacetamide (9 mM). Proteins were digested on-filter (16 h, 37 °C) with sequencing-grade modified trypsin (50:1, Promega).

**Measurement of protein turnover and steady-state abundance.** Protein turnover and abundance measurements were performed as described in detail in the open data descriptor[25]. Briefly, mouse plasma was analyzed with gas chromatography mass spectrometry. Plasma sample from each time point (20 μL) was mixed with NaOH (10 N, 2 μL) and acetone (5% v/v, 5 μL) in acetonitrile. Standard curves were created by mixing to the acetone 0–20% molar ratio of $^2H_2O$ at 11 regular intervals in phosphate-buffered saline in lieu of mouse plasma sample. The sample mixtures were incubated at ambient temperature overnight. Acetone was extracted by adding chloroform (500 μL) and anhydrous sodium sulfate (500 mg). The extracted solution (1 μL) was analyzed on a gas chromatography mass spectrometer (Agilent 6890/5975) with a J&W DB17-MS capillary column (Agilent, 30 m × 0.25 mm × 0.25 μm) at the UCLA Molecular Instrumentation Center. The column temperature gradient was as follows: 60 °C initial, 20 °C·min⁻¹ increase to 100 °C, 50 °C min⁻¹ increase to 220 °C, 1 min hold. The mass spectrometer operated in the electron impact mode (70 eV) and selective ion monitoring at $m/z$ 58 and 59 with 10 ms dwell time.

First-dimension (high pH) separation was conducted by resolving peptides on a Phenomenex C18 column (Jupiter Proteo C12, 4 μm particle, 90 Å pore, 100 mm × 1 mm dimension) at high pH using a Finnigan Surveyor liquid chromatography system[62]. The solvent gradient was as follows: 0–2 min, 0–5% B; 3–32 min, 5–35% B; 32–37 min, 80% B; 50 μL min⁻¹; A: 20 mM ammonium formate, pH 10; B: 20 mM ammonium formate, 90% acetonitrile, pH 10. Fifty micrograms of proteolytic peptides were injected with a syringe into a manual 6-port/2-position switch valve. Twelve fractions from 16–40 min were collected, lyophilized and re-dissolved in 20 μL 0.5% formic acid with 2% acetonitrile prior to low-pH reversed-phase separation. On-line second-dimension (low-pH) reversed-phase chromatography was performed on all samples using an Easy-nLC 1000 nano-UPLC system (Thermo Scientific) on an EasySpray C18 column (PepMap, 3 μm particle, 100-Å pore; 75 μm × 150 mm dimension; Thermo Scientific) held at 50 °C. The solvent gradient was 0–110 min: 0–40% B; 110–117 min: 40–80% B; 117–120 min: 80% B; 300 nL min⁻¹; A: 0.1% formic acid, 2% acetonitrile; B: 0.1% formic acid, 80% acetonitrile. Each high pH fraction was injected (10 μL) by the autosampler to the Easy-nLC 1000 nano-UPLC system.

High-resolution Fourier-transform tandem mass spectrometry was performed on an LTQ Orbitrap Elite instrument (Thermo Scientific), coupled on-line to an Easy-nLC 1000 nano-UPLC system (Thermo Scientific) through a Thermo EasySpray interface. Signals were acquired in FT/IT mode: each FT MS1 survey scan was analyzed at 60,000 resolving power in profile mode, followed by IT MS2 scans on the top 15 ions. MS1 and MS2 target ion accumulation targets were 1.0E4 and 1.0E6, respectively. MS1 lock mass ($m/z$ 425.120025) and dynamic exclusion (90 s) were used. Peptide identification was performed using the database search algorithm ProLuCID[63] against a reverse-decoyed protein sequence database (Uniprot Reference Proteome, reviewed, accessed April-08–2014, 16,672 forward entries). Static cysteine carboxyamidomethylation (57.02146 Da) and ≤3 of the following variable modifications were allowed: methionine oxidation (15.9949 Da), lysine acetylation (42.0106 Da), serine/threonine/tyrosine phosphorylation (79.9663 Da), lysine ubiquitylation (114.0403 Da), and asparagine deamidation (0.9840 Da). Tryptic, semi-tryptic, and non-tryptic peptides within a 20-ppm mass window surrounding the candidate precursor mass were searched. Protein inference was performed using DTASelect[64] (v.2.0), requiring ≤1% global peptide false discovery rate and 2 unique peptides per protein. Modified or non-tryptic peptides were subjected to separate statistical filters to limit false discovery[64].

Mass spectra were converted into mzML format[65] using ProteoWizard[66] (v.2.1), then input for analysis in ProTurn[21] (v.2.0.5). ProTurn retrieved identified peptides that were uniquely assigned to proteins for area integration. Extracted ion chromatographs were processed with Savitzky-Golay filters, and the areas-under-curves within 60 ppm of the peptide mass at the retention time of the identified peptides were integrated. Peptides that were explicitly identified and integrated in ≥4 time points were accepted for the calculation of protein abundance and turnover. The areas-under-curve of up to the first six peptide isotopomers for each qualifying peptide at each time point are integrated over retention time space for abundance and turnover calculation. To calculate abundance, the summed areas of all isotopomers within each peptide envelope were normalized against total spectral intensity, then normalized against the number of possible tryptic peptides (6–30 amino acids) from in silico digestion of the protein. Abundance changes are presented as $\log_2$ ratios of abundance at each hypertrophy time points over control day 0 samples. To calculate turnover rates, the fractional abundance of the first mass isotopomer from each integrated time point ($A_0$) was modeled with a compound kinetic curve and the best-fit parameter was optimized using the Nelder-Mead algorithm. The $A_0$ values measured by the mass spectrometer from each of seven time-points served as independent biological replicates to allow estimation of the best-fitted values as well as variance of the turnover rate constant

($k$). Only a subset of peptide isotopomer time-series with goodness-of-fit ≥0.8 or standard error of estimate ≤0.05 were accepted for turnover calculation[25]. Protein-level turnover rate was reported as the median and median absolute deviation of the best-fitted turnover rate constant of each accepted constituent peptide that is unique in the proteome. Protein turnover flux is reported as the product between the turnover rate constant and fractional abundance[23].

Tandem mass tags (10-plex) were purchased from Thermo Scientific. Peptides were labeled according to manufacturer's instruction. The TMT10plex quantification method within Proteome Discoverer software was used to calculate the reporter ratios with mass tolerance ±10 ppm. Only peptide spectra containing all reporter ions were designated as quantifiable. Protein ratio was expressed as a median value of the ratios for all quantifiable spectra of the peptides pertaining to that protein.

**Functional annotations and data sources.** Protein annotation and functional category overrepresentation analyses were performed by fetching gene ontology from the Ensembl database on June 18th 2015 using the biomaRt package in R (v.3.2.1)/Bioconductor[67]. In all functional enrichment analyses, the list of total proteins whose abundance and turnover were quantified in a sample was used as the comparison background. We retrieved all protein–protein interaction pairs concerning the quantified proteins in the omics data sets from three curated molecular interaction databases (IntAct, BioGrid, and MintDB)[68]. To avoid bias, we removed all self-interactions and calculated all pairwise absolute fold-change of protein–protein interacting pairs. Permutation of interaction networks was done via label exchange by randomly sampling without replacement each of all curated interacting partners with quantified turnover rates then calculating the absolute distance in $\log_2$ turnover rates between each pair. Average numbers of interactions among proteins in each category (slow, medium, and fast turnover) after 10 million repeated permutations were reported. Protein association data were queried from STRING database v.10 *Mus musculus* entries[55]. Cellular pathway information was retrieved from Reactome (release V53, July 2015), where a pathway group is defined as a set of pathways retrieved from Reactome that completely or partially overlap by ≥50% and that contain ≥3 pathway members among the implicated disease proteins.

Public transcriptomics data sets were retrieved from Gene Expression Omnibus on January 26th, 2016 using the search terms "heart" and "hypertrophy". Data were fetched using the *GEOQuery* package[69] getGEO API as implemented on R/Bioconductor. Inclusion criteria of the data set include the experimental model being that of pathological hypertrophy of the heart, induced by surgical or pharmacological means (transverse aortic constriction, myocardial infarct, isoproterenol, or angiotensin II) for one or more days on adult animals. Data from male and female wild-type animals were included. Data from genetic models of pathological hypertrophy or exercise-induced physiological hypertrophy are excluded. Single-channel array data were used. Data from exon arrays, data without replicates, or those with abnormal quantile-quantile profiles, are excluded. Data whose probe IDs cannot map to UniProt entries are excluded. Included data series were as follows: GSE76, GSE1621, GSE2459, GSE5500, GSE6970, GSE7487, GSE12337, GSE18224, GSE18801, GSE24489, GSE27689, GSE29145, GSE30314, GSE37597, GSE47420, GSE48760, GSE56348, GSE69355[28,33–47]. Not every experiment within each data series was included as some data series contained experimental data from both wild-type and genetically modified animals. To avoid platform and batch bias, all data sets were first analyzed separately to compare the standardized fold-change between hypertrophic and normal hearts. Log fold-change of RNA data in the meta-analysis was estimated using a linear model as implemented in the *limma* package in R (v.3.2.1)/Bioconductor[70].

**Statistical analysis.** Significance in changes of protein turnover or abundance was reported as $P$ of Wilcoxon's test between two samples, or Kruskal–Wallis test with Dunn's post hoc analysis amongst multiple samples. A protein is considered consistently differentially regulated if it shows 80th percentile or above changes in ≥4 mouse strains, and has corresponding significance of $P < 0.05$ from a combined Stouffer's Z score test. Significance of correlation was reported as asymptotic $P$ of Spearman's $\rho$. The similarity of gene lists in the meta-analysis of multi-omics data sets was compared using the *OrderedList* package[48] (v.2.4) in R/Bioconductor. Significance of functional enrichment was reported as hypergeometric test $P$ with Benjamini–Hochberg adjustment. The significance of protein–protein correlations and protein–trait correlations was reported as the asymptotic $P$ value of Spearman's $\rho$. The significance of functional enrichment was reported as the $P$ value of hypergeometric test, followed by Benjamini–Hochberg adjustment for false discovery rates in testing multiple hypotheses. Results with raw $P$ value < 0.05 before false discovery adjustment are considered suggestive. Results with $P$ values < 0.05 after false discovery adjustment are considered significant. Clustering of coexpression and co-turnover data were performed with the aid of the Weighted Gene Correlation Network Analysis (WGCNA) package[49] in R v.3.2.1.

**Code availability.** The processed data and data processing code are available on Synapse (https://doi.org/10.7303/syn2289125). Software ProTurn (v.2.0.5) is freely available at http://heartproteome.org.

**Data availability**. All mass spectrometry data are publicly available on ProteomeXchange (accession PXD002870). The processed data are available on Synapse (https://doi.org/10.7303/syn2289125) and figshare (https://doi.org/10.6084/m9.figshare.c.2171334).

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

## Acknowledgements

This study was supported in part by US National Institutes of Health funding R35-HL135772, and U54-GM114833 to P.P.; R00-HL127302 to M.P.Y.L.; and F32-HL-139045 to E.L.

## Author contributions

E.L.: Contributed to study design, data analysis, and manuscript writing. Q.C.: Contributed to animal experimental model and data acquisition. M.P.Y.L.: Contributed to study design and data acquisition. J.W.: Contributed to the data acquisition. D.C.M.N.: Contributed to the data acquisition. B.J.B.: Contributed to software programming. J.M.L.: Contributed to critical revisions of the manuscript. D.A.L.: Contributed to animal experimental model. D.W.: Contributed to animal experimental model. H.H.: Contributed to the data analysis. P.P.: Contributed to study design, funding acquisition, and manuscript writing.

## Additional information

**Competing interests:** The authors declare no competing financial interests.

