## [Peer Review File · Nature Communications]

Reviewers' comments:

Reviewer #1 (Remarks to the Author):

This is something of a tour de force in scope and scale. The work, looking at dynamics, protein:transcript relationships and protein abundances. By my calculations, this study would have taken at least a year of data acquisition, let alone all of the prior experimental design/sample prep and subsequent data analysis.

I maintain an active and collegiate dialogue with this group, and my comments below should be seen as being supportive, to make the paper as good as it can be.

I maintain that plotting k values in log space may meet the condition of a log normal distribution of values (un surprising) but the goal is not to generate a 'Gaussian' distribution. It should be to actually show what the tissues are doing. Looking at the experimental design, and analytical approach, I think the actually measurable dynamic range is between about $0.02/d$ and $8/d$ - the current axis limits imply that values were obtained that exceeded these boundaries - these have to be delivered by few time points and will be error prone. Some discussion about this issue would be welcome. The $\log(2)$ plots of k range from 2^{-6} to 2^{+4} ...10 logs of dynamic range - is this feasible with the sampling protocol here (first point 1d, last point 14d)?

Figure 1: I strongly recommend that panel A be redone. It implies that isoP treatment preceded D2O administration, but on reading the ms, we discover that the isoP and the D2O were started at the same time. This discovery now makes me want to ask some deeper questions.

In particular, since isoP induces hypertrophy (rather than hyperplasia) the incorporation of label (which leads to the turnover parameter) is more complex, as it reflects incorporation of new protein as the pool expands, and the turnover of the existing pool. My reading of Supp T1 suggest that the HW/BW ratio (I assume this is percent, it is not defined) moves from ca 4% to ca 6% - in other words, a 50% expansion of the tissue biomass, and thus, will show a major incorporation of label due to additional synthesis. How was pool expansion corrected for?

The main reason to address this is because the correlation between turnover and transcript may be compromised by the conflation of two parameters (turnover and growth).

Related: we're not given the body weight changes of the mice. They are young adult, but could still be growing - thus whilst HW:BW increases, if BW is increasing as well, the pool expansion is even greater. This should be addressed with clearer stats and data analysis, as well as in the paper itself.

The protein:transcript comparisons are interesting, and unsurprising, but I will query the use of spectral counts as the surrogate for abundance. I might suggest that MS1 intensity based LFQ is now sufficiently well recognised as a better measure of abundance than spectral counts that these should be used. It is actually rather hard to discover how the LFQ values were obtained - given that the isotopomer profiles are changing during labelling and this might alter the selection of precursor ions - were common ions used only? hard to see how with SC, but OK with intensity MS1 based LFQ.

In several places in the manuscript, the authors discuss the relationship between transcript and

turnover, but in reality, the connectivity should be between transcript and synthesis (not the same as turnover, when $dP/dt = \text{growth}$, is non-zero).

There could well be value in including a measure of flux instead of rate constants, by multiple rates with label-free estimates of pool magnitude. It is possible that an increase in pool size could masquerade as an apparent increase in turnover. The recent Hammond MCP paper is an example of how this can be used. Also, this particular paper is a clear demonstration of the issues associated with using cells in culture as any indication of what is going on in tissues!

Lastly, I do not believe that enough is made of the remarkable strain differences. The B6 mice show about a 4% HW:BW, the DBA 6.5%. But, what matters are the values at the start and end of the experiment - the changes in HW:BW are the important parameter - 109% in B6, 128% in DBA - are these changes significantly different. The DBA proteins should thus show greater labelling due to hypertrophy.

Overall (significant changes needed)

Simplify the arguments and assert the hypotheses more clearly. There are a lot of different analyses in here, but it is not that easy to work out their relatedness. It might be prudent to ask if some part of this work could be published elsewhere. What are the most important Q: transcript/ks relationships, pool size/flux balance? Growth corrected true increase in synthesis.

I am quite unpersuaded by the log2 transform of kdeg values, which a) are limited in dynamic range, and the linear parameter. The compression of values might look nice, but the graphs are for the reader, and this is a distortion of the true behaviour.

Sup T1 - add animal weights

Fig 1: simplify and clarify experimental design image - isoP starts at same time as D2O - this is actually important - see growth comments

Figure 4A: Showing the HW:BW ratio (again, clarify this is %) is a little confusing, because, for example the DBA hearts are a higher % at the start - thus, the change is critical, rather than the actual proportion

Figure 4C/D - there are six strains, and one measure, so why are their six bold points and six faint points - not explained in legend, and not easy to understand. The dark symbols actually cluster tightly with little evidence of a trend, so the faint symbols are important to the conclusion implied by the correlation diagrams. again, I would prefer to see the log scaling removed - the dynamic range of kdeg values is small (by definition, since the labelling window is short)

All figures that plot kdeg (is it really kdeg, if there is growth?) should indicate the dimensions (d). Cell biologists would probably interpret those numbers as h!

Supp F1, is spectral counting as good as is needed for this analysis? - please reconsider

Supp F2, data would be better if we have absolute rates and errors on kdeg measure..

Explain how the LFQ data were obtained - which samples? which time point in labelling? separate experiment at $t=0$?

Manuscript

Make clear when the ms is talking about turnover (actually controlled by kdeg) rather than synthesis

Clarify the compensation data correction of ks for growth

The animals were taken at the same time, and kept on a 12h light/dark cycle - but we are not told where in the activity/dark cycle the animals were - please explain
This reviewer had never heard of inter-vigintile range, and neither has Google! (it threw up one paper, plus translations!). Please explain this term, or correct to what was meant.

I do hope the authors will be able to address these questions.
With my best wishes
Rob Beynon

Reviewer #2 (Remarks to the Author):

Heart remodeling is an important physiological and pathological process whose regulation is still poorly understood- especially unclear is how the temporal dynamics of thousands of transcripts and proteins in many cell types (only ~50% of the heart cells consists of cardiomyocytes) is coordinated.

Lau et al use a combination of transcriptomic and cutting-edge proteomic methods and data mining to analyze the changes in transcripts and protein levels during isoprenaline-induced hypertrophic remodeling, one of several commonly used hypertrophy models.

The comprehensive use of six established mouse genetic lines and their comparative analysis is a particular strength of this work.

However, it seems a bit overblown to emphasize a "hitherto unknown association between metabolic remodeling in hypertrophy and coordinated protein turnover in selected sub-proteomes". Recent work e.g. on the coordinated turnover of myofibrillar proteins, metabolic enzymes and transcription factors by the muscle-specific TRIM ubiquitin E3 ligases MURF1/2/3 showed clearly that groups of cardiac protein turnover regulators target selected sub-proteomes- it would be useful to refer to some of these data and highlight changes of these established turnover regulators briefly, even if only in the supplements. Nevertheless, the comprehensive, not target-gene focused approach here reveals a large haul of novel data that will be valuable for cardiac research. Overall, this is an interesting, rigorously performed analysis that yields large amounts of novel data and insight into cardiac remodeling.

While the resulting data are surely interesting, they very difficult to penetrate. 70 candidate disease molecules were identified, many of which apparently previously not associated with remodeling-associated turnover changes, but these results are somewhat forlorn in the supplemental tables and their functional context must be pieced together.

While this paper will be of immediate interest to the proteomics community, in its current form the figures and tables are of considerable complexity to the point of being almost impenetrable. Quite generally, the paper would significantly benefit from a more accessible presentation of the results and the key findings.

For example:

surprisingly, sarcomeric proteins, which are recognized to undergo major changes during remodeling due to altered isogene expression, differential splicing and altered degradation kinetics do not feature recognizably. While the discussion alludes to contractile and sarcolemmal proteins, these are missing as a recognizable category in most figures. For most cardiovascular scientists, these would clearly be benchmark changes to put the data concerning other cellular pathways as well as cell types into perspective. Would it be possible to add contractile and sarcolemmal proteins to Figure 3 D and Supplemental Figure S3?

It is not necessarily surprising, but unprecedented in the depth of the analysis presented here, to see the turnover of protein-protein interacting partners being coordinated. Could exemplary pairs for the main functional groups be highlighted.

Supplemental Table S2 Disease_Prots shows truncated protein names in column 2 (PN), and would significantly benefit from a much clearer layout, complete protein names, and more explanation. This is after all one of the key results, identifying the 70 novel targets of cardiac remodeling. The table entries are incomprehensible for anyone but a mass spectrometry expert- what is FALSE and TRUE and what does it tell the reader about the data?

Columns "Group" and "Rep" are truncated in table Supplemental Table S4. What are the red highlights in this table?

Line numbers in table descriptions in the supplements do not match up with the table referred to.

Minor comments

Some smaller grammatical mistakes (e.g. ubiquitin in Fig 3D) and labels on subfigures missing: S3, D missing.

Reviewer #3 (Remarks to the Author):

The manuscript from Lau and co-workers used an isoproterenol-induced hypertrophy model to investigate the overlap of mRNA levels, protein abundance, and protein turnover. For the analysis of protein turnover rates the authors used deuterium levels and a previously developed model to calculate absolute turnover rates. By overlaying three independent datasets the authors potentially identified 70 proteins with a disease related function. Overall the systematic analysis of protein turnover rates and protein abundance is an important new field and opens a new direction how proteins are regulated during any perturbation. However, the manuscript is very descriptive and absolute no functional analysis were performed.

Major points:

The authors should explain the workflow proteins abundance were measured in the results part. It is not clear why the authors identified more than 8000 proteins and quantified only 3200 proteins. What happens with the other 5000 proteins. In addition, the comparison to the mRNA dataset revealed only 1900 overlapping proteins. One should expect that each protein is at least expressed and present in the mRNA database. The authors stated that the overlap of protein turnover and protein abundance is poorly correlated but did not explain/discuss this inconsistency. The authors selected six mouse strains and it seems that each strain were analyzed only once without any biological replication. It would be nice to have at least for some mouse strains a biological triplicate analysis to judge the significance of the dataset. The cut-off seems to be arbitrary. Why have the authors selected a cut-off of 1.4 for increased turnover and decreased turnover 1.2. In the case the authors performed biological replicates one should perform a FDR analysis based permutation based statistics. The chapter on page 3 (unfortunately no page numbers are present) line 23-36 is difficult to understand and should be rephrased. For example "These proteins are..." which one? And "They preferentially function..."

The authors stated a "modest overlap" please quantify this. It might be useful to calculate the AUC (area under the curve) for some proteins with matching and non-matching protein abundance and turnover. Unfortunately, the first part is difficult to understand and the overlap of all proteins between different conditions is definitely very confusing for readers which are not in the proteomics field. The authors identified 70 proteins with altered turnover. What is the proportion of enhanced and decreased turnover here? What is the significance of this candidates? The authors

described that DHRS7C as a protein with decreased protein turnover but increased protein abundance. It would be nice to confirm this by an alternative approach and it be that the ubiquitination is also affected to stabilize the protein. Otherwise this part remains quite speculative. The authors found increased mRNA levels of "inflammatory response". This might reflect the presence of other cell types such as macrophages. Overall the discussion fails to explain the reason for the divergent regulation of protein abundance and turnover. In addition, the combination of six mouse strains and the hypertrophy model seems to be not very helpful to identify new proteins regulated by hypertrophy. A more focused analysis of one strain (such as in the mRNA datasets) might be sufficient to describe the effect and reduces the complexity of the presented data.

Minor points:

Usually, histones are very stable proteins. Which half-life or turnover rate did were measured for HIST1H4A? Have the authors used only unique peptides for this histone? It might be that a prolonged labelling time (more than two weeks) will give much sharper results.

What is the mRNA level of DHRS7C in the mRNA dataset? It might be helpful to integrate it in Fig. 1E.

"Rank aggregation algorithm" should be explained shortly in order to allow the reader to follow the subsequent section.

Figure 3B and line 27, page 6: "We observed that Type I and Type II proteins differed from Type III proteins in that they exhibited greater expression changes during hypertrophy"

It is hard to observe this statement in Figure 3B. Another representation might be more appropriate. Is the difference significant?

The permutation tests for interactome analysis are not described. Figure 5E,F are not mentioned in the text. Page 8, line 35: Figure5E should be Figure 5G

In general, the axis titles are often not very clear, i.e. it should always be indicated on the axis which ratio or which turnover rate is shown e.g. hypertrophic/normal.

Figure 2D: Do not demonstrate the protein abundances high rank similarities between themselves?

Suppl. Fig.2A: It gets not clear if for left and right plot the same amount of proteins was used. Relative Frequency instead of absolute frequency should be used.

Suppl. Fig. 2B: Only one replicate was measured per mouse strain. Shouldn't there be more replicates per strain to draw conclusions about strain specific turnover. Couldn't be the ratios just be observed by chance?

Suppl. Fig. 2C: Figure is too small and individual proteins are not readable.

Suppl. Fig.3: "D" is missing in the labelling.

Reviewers' comments:

Reviewer #1 (Remarks to the Author):

Comment 1: This is something of a tour de force in scope and scale. The work, looking at dynamics, protein:transcript relationships and protein abundances. By my calculations, this study would have taken at least a year of data acquisition, let alone all of the prior experimental design/sample prep and subsequent data analysis.

I maintain an active and collegiate dialogue with this group, and my comments below should be seen as being supportive, to make the paper as good as it can be.

Response: We are grateful to the Reviewer for a thorough evaluation of our manuscript and for the many helpful comments. We are especially encouraged by the Reviewer's appraisal on the scope of the dataset.

Comment 2: I maintain that plotting k values in log space may meet the condition of a log normal distribution of values (unsurprising) but the goal is not to generate a 'Gaussian' distribution. It should be to actually show what the tissues are doing. Looking at the experimental design, and analytical approach, I think the actually measurable dynamic range is between about $0.02/d$ and $8/d$ - the current axis limits imply that values were obtained that exceeded these boundaries - these have to be delivered by few time points and will be error prone. Some discussion about this issue would be welcome. The $\log(2)$ plots of k range from 2^{-6} to 2^{+4} ...10 logs of dynamic range - is this feasible with the sampling protocol here (first point 1d, last point 14d)?

Response: We thank the Reviewer for this insight. We agree that the choice of logarithmic vs. linear scale can impact data interpretation from the readers, and that in some instances there are clear advantages to presenting data in linear scale, e.g., such plots can highlight the existence of fast-turnover proteins with very short half-life, which may contribute preponderantly to the total protein turnover flux of an organ. Following the Reviewer's comment, we have included linear-scale figures where appropriate, including Figures 5C, 5D; and Supplementary Figures 1E, 5A, to show the distribution of measured protein turnover rates in linear scale. Please also see our response to Comment 11 below for our rationale for casting turnover rate data in logarithmic scales.

Secondly, we agree with the Reviewer that the experimental sampling time points would place a limit on the range of turnover rates that may be accurately derived from the data. In our prior data descriptor article we described our rationale to optimize filtering criteria based on inter-peptide variance without any ad hoc supervision. In our method, relatively few well-fitted isotopomer data series (0.9%; 1172 among 120,454) from the dataset exceed the range between 0.02 and $4 d^{-1}$, which falls within the timeframe where turnover rate constants might be accurately deduced by isotopomer data at the sampling time points. The proportion of extreme values is further lower at the protein level and when considering only the 1,901 reliably quantified proteins that form the basis of analysis in the manuscript: only 0.7% of compared proteins (14 out of 1901) have average turnover rates outside the 0.02 to $4 d^{-1}$ range. In rare instances where the curve-fitting optimization does not converge, we have eschewed ad hoc data removal, and we believe that these data points are in minority scenarios and they possess

nondemonstrable impact on the overall analysis. Certainly a lengthening of duration of labeling would undoubtedly expand the range of turnover rates, however, experimental challenges also manifest in the exorbitant number of samples that must be analyzed (6 strains x 2 conditions x 7 time points x 12 fractions). We have previously labeled mice with identical doses of deuterium oxide as employed here for up to 90 days (Kim et al., 2012), and found that the majority of cardiac proteins fall within the current experimental labeling range (median half-life of proteins in the heart is ~7 days).

Comment 3: Figure 1: I strongly recommend that panel A be redone. It implies that isoP treatment preceded D2O administration, but on reading the ms, we discover that the isoP and the D2O were started at the same time. This discovery now makes me want to ask some deeper questions.

Response: We thank the Reviewer for this comment. We have now revised Figure 1A to more clearly illustrate that deuterium administration occurs simultaneously to the induction of cardiac hypertrophy. Specifically in the schematic we now separate the biological factors (strain vs. hypertrophy) from the workflow, and explicitly state that deuterium administration is concurrent to isoproterenol.

Figure 1A: Schematic of multi-omics analysis.

Comment 4: In particular, since isoP induces hypertrophy (rather than hyperplasia) the incorporation of label (which leads to the turnover parameter) is more complex, as it reflects incorporation of new protein as the pool expands, and the turnover of the existing pool. My reading of Supp T1 suggest that the HW/BW ratio (I assume this is percent, it is not defined) moves from ca 4% to ca 6% - in other words, a 50% expansion of the tissue biomass, and thus,

will show a major incorporation of label due to additional synthesis. How was pool expansion corrected for?

The main reason to address this is because the correlation between turnover and transcript may be compromised by the conflation of two parameters (turnover and growth).

Response: We thank the Reviewer for this comment. We now clearly define HW/BW (milligrams in heart weight per gram of total animal body weight) throughout the text and in the figures, including in the axis labels of Figures 5C, 5D; and Supplementary Figures 6A, 6B, 7F, 7H. To account for pool expansion in the current study, we assume the empirically measurable rate constant k to reflect an unobserved mixture of synthesis and degradation processes. In other words, the turnover rate constant is defined here as the rate constant in relation to the time at which a fixed proportion of the protein pool becomes replaced with new proteins, as opposed to absolute turnover (flux of molecules through protein pool per unit time). As the Reviewer is aware, in non-constant pool sizes the measured turnover rate is a conflation of contributions from protein synthesis rate and protein degradation rate, which cannot be assumed as being equal to each other. Although the overall protein pool may expand for some proteins, they may expand via increase in synthesis or decrease in degradation, which should result in different empirical turnover rate. Corroborating this point, we now show that under proteasomal inhibition by epoxomicin, 83% of measured protein show decreased turnover rates (Supplementary Figure 4), indicating the measured turnover rates are responsive to protein degradation processes. Hence to avoid potential confusion, we eschew notations of protein synthesis rate constant (k_{syn}) or degradation rate constant (k_{deg}) in the manuscript.

We believe that the ability to preserve isotope enrichment patterns in changing pool sizes confers a major advantage on measuring protein turnover as a means to discover gene regulatory signatures. Because the signatures of protein induction are maintained in self-normalized isotopic ratios, they are not diluted by sample loading normalization per unit heart weight. In other words, if the majority of protein species in the heart were to increase in expression by the same proportion in a concentrically hypertrophic heart, a conventional proteomics experiment would be unlikely to detect such differential expression due to the effect of loading normalization.

Lastly, we note that we have employed identical models of micro-osmotic pump-delivered isoproterenol to initiate cardiac hypertrophy in multiple previous studies (see also (Lam et al., 2014; Lau et al., 2016)), as well as examined the gradual recovery following isoproterenol withdrawal. A key observation that informed our choice of the isoproterenol model was its graduality in inducing protein turnover changes. In our experience, similar numbers of peptide ion time series were well-fitted to a two-compartment kinetic model between normal and hypertrophic hearts, suggesting that the majority of protein turnover changes are not due to sudden changes in protein production or pool size.

Comment 5: Related: we're not given the body weight changes of the mice. They are young adult, but could still be growing - thus whilst HW:BW increases, if BW is increasing as well, the pool expansion is even greater. This should be addressed with clearer stats and data analysis, as well as in the paper itself.

Response: We thank the Reviewer for this comment. During our experiments, we have observed fluctuations in mouse body weights, possibly due to random variations in food/water intake. However, there has not been a trend indicating overall growths of the mice. Inbred mice between 9 to 12 weeks of age are frequently employed in cardiac hypertrophy experiments in the cardiovascular literature, and as the Reviewer suggests they are considered to be young adults. We have observed no evidence of increase in body weight in the animals over the

experimental periods. The body weights of animals at each experimental time points are shown below for the Reviewer's consideration (see Figure R1 below).

R1

Figure R1. Mouse body weight. Body weights (in grams) of mice in six genetic backgrounds at each time point (in days) in the experiment are shown. Box: 25-75th percentile; Whiskers: 5th-95th percentile. (n = 53 for A/J; 48 for BALB/cJ; 23 for C57BL/6J; 51 for CE/J; 48 for DBA/2J; 17 for FVB/NJ.)

Following the Reviewer's suggestion, we also employed a general linear model ($BW \sim \text{time} + \text{strain} + \text{hypertrophy} + \text{time} \times \text{hypertrophy}$) to examine whether the experimental time points (d) may influence body weight (g) given strain and disease status as covariates. We found that whereas body weight is, expectedly, significantly influenced by animal strain (vs. A/J: BALB/cJ $P=2.3e-14$; C57BL/6J $P<2e-16$; CE/J $P<2e-16$; DBA/2J $P=8.3e-9$; FVB/NJ $P<2e-16$), neither the experimental time point ($P=0.26$), nor their disease status ($P=0.41$), nor their interaction term ($P=0.92$), was found to significantly influence body weight in the model.

Thirdly, we note that while it is possible to perform observational analyses using gross weights of the heart or other organs, as our collaborators have successfully done for the HMDP panel of 100+ strains of recombinant inbred mice for their heart and liver weights, prior to data collection we have elected to measure HW/BW because it is a well-accepted metric of cardiac hypertrophy in the cardiovascular research literature (Heineke et al., 2010; Hill et al., 2002; Rockman et al., 1991). We are of the view that the common rationale for reporting HW/BW (or, less commonly, HW/T heart-weight-to-tibia-length) critically lessens variance by normalizing against isometric variations in body sizes within mouse cardiac hypertrophy models. HW/BW also correlates well with echocardiographic measures of cardiac hypertrophy including diastolic volume (Supplementary Figure 7A, see also (Lam et al., 2014)), likewise corroborating that HW/BW ratio serves as a valid measure of cardiac hypertrophy within the current experimental design.

Finally, we note that our experimental design does not mandate samples at individual time points from biological replicate groups to be collected in chronological sequence, hence hypothetical differences in growth rates are more likely to contribute to variance but not bias in measured isotope incorporation rates. Following the Reviewer's comment, we have now clarified this point and included the mouse strain's body weight data in Supplementary Table 2 in the revised manuscript.

Comment 6: The protein:transcript comparisons are interesting, and unsurprising, but I will query the use of spectral counts as the surrogate for abundance. I might suggest that MS1 intensity based LFQ is now sufficiently well recognised as a better measure of abundance than

spectral counts that these should be used. It is actually rather hard to discover how the LFQ values were obtained - given that the isotopomer profiles are changing during labelling and this might alter the selection of precursor ions - were common ions used only? hard to see how with SC, but OK with intensity MS1 based LFQ.

Response: We thank the Reviewer for this comment. We have now performed additional analysis to show comparable data distributions from spectral counting vs. intensity-based approaches in (Supplementary Figure 1A-B of the revised manuscript). We found general agreement between chromatographic and spectral count based measures although variations exist. However, we hope the Reviewer will agree that a comprehensive evaluation of the two methods is beyond the scope of the current dataset and manuscript. Both are commonly to measure protein expression in quantitative proteomics experiments, e.g., in two draft maps for the human proteome (for LFQ e.g., (Wilhelm et al., 2014); spectral count e.g., (Kim et al., 2014)).

Prompted by the Reviewer's comment we have now clarified the method to obtain MS1-based label free quantitation (LFQ) values under changing isotopomer profiles in the Methods section. Our software pipeline ProTurn allows simultaneous calculation of protein relative abundance and protein turnover under a common computational pipeline. To get MS1-based LFQ for each peptide, we take the sum of intensities of all individual isotopomers, which are integrated for both abundance and turnover calculation (see also (Lau et al., 2016)). For the purpose of spectral counting, we found from labeled control animals that isotopomer shifting is not a major concern for the search engine (ProLuCID) we employ, which is instructed to take into account +1 and +2 peaks in all peptide-spectrum matches. We now describe the method of chromatographic peak intensity-based label-free quantification (MS1 LFQ) calculation in detail:

Page 15 lines 21-27 *"The areas-under-curve of up to the first six peptide isotopomers for each qualifying peptide at each time point are integrated over retention time space for abundance and turnover calculation. To calculate abundance, the summed areas of all isotopomers within each peptide envelope were normalized against total spectral intensity, then normalized against the number of possible tryptic peptides (6–30 amino acids) from in silico digestion of the protein. Abundance changes are presented as \log_2 ratios of abundance at each hypertrophy time points over control day 0 samples."*

Moreover, we now provide additional protein quantification data based on tandem mass tag (TMT) labeling in C57BL6/J mice (Supplementary Figure 3). Protein turnover signatures show poor overlap with the stable isotope labeling quantification data. The results are consistent protein turnover signatures being not captured by labeled quantification results, and corroborate our conclusion that multi-omics data support the discovery independent disease signatures in cardiac hypertrophy. Please also see response to Reviewer 1 Comments 17 and 19.

Supplementary Figure 3. Protein quantification in hypertrophic mouse hearts using tandem mass tags. A: Schematic of tandem mass tag (TMT) experiments. Triplicate groups of C57BL/6J mice were administered with 15 mg/kg/d isoproterenol via microosmotic pumps for up to 14 days. Cardiac proteins were collected at day 0, day 7, day 14 and labeled with 10-plex TMT labels. Each set of TMT-labeled samples (9 channels) were analyzed in technical triplicates. B-C: Volcano plots of $-\log_{10} P$ value over \log_2 fold change in (B) day 14 of hypertrophy and (C) day 7 of hypertrophy against control sample. D: Heat map of differentially regulated proteins in TMT experiments at $FDR < 0.05$ and absolute \log_2 fold-change of 0.5. Right: Only four of the differentially regulated samples were also consistent turnover signatures; only three were consistent RNA signatures. E: Upon relaxing the TMT signature criteria ($FDR < 0.05$; any magnitude), only 17 of 99 turnover signatures and 20 out of 110 RNA signatures show differential regulation at the protein abundance level. Hence taken together the data do not support that an orthogonal method of protein quantification is able to recapitulate turnover and RNA signatures.

Comment 7: In several places in the manuscript, the authors discuss the relationship between transcript and turnover, but in reality, the connectivity should be between transcript and synthesis (not the same as turnover, when $dP/dt = \text{growth}$, is non-zero).

Response: We thank the Reviewer for raising this point. Herein we employ the term turnover to account for the combination of synthesis and degradation processes that cause the proportional replacement of a protein pool at characteristic rates. The Reviewer is correct to point out that in the described system dP/dt is likely to be non-zero in many protein pools during cardiac hypertrophy, where there is an overall increase in cardiac mass. We have also noted this observation in a previous publication, and discussed potential scenarios wherein changes in protein abundance may be uncoupled from changes in protein turnover (Lam et al., 2014). Following the Reviewer’s comment we have now included the changing pool size as a potential future study in the Discussion section:

Page 11 lines 33-35 *“Future work may also exploit post-translational dynamics to fine-map mechanisms of gene regulation in individual cell types, or to employ absolute quantification and modeling to distinguish synthesis and degradation rates.”*

Please also see our response to Reviewer 1 Comment 4.

Comment 8: There could well be value in including a measure of flux instead of rate constants, by multiple rates with label-free estimates of pool magnitude. It is possible that an increase in pool size could masquerade as an apparent increase in turnover. The recent Hammond MCP paper is an example of how this can be used. Also, this particular paper is a clear demonstration of the issues associated with using cells in culture as any indication of what is going on in tissues!

Response: We appreciate the Reviewer’s suggestion. In response we have performed two new analyses. We have included basal flux information for the 273 candidate signature proteins in the Supplementary Table 3 for reference, calculated using the methods of (Hammond et al., 2016). Secondly, we now compare protein flux with protein abundance and turnover in Supplementary Figure 2. As the Reviewer points out and as demonstrated in (Hammond et al., 2016), total protein flux is likely to be influenced heavily by protein abundance, which has a wide dynamic range than the typically measured turnover rate constants. We observed no linear relationship between turnover log ratios in hypertrophy vs. basal flux. Details of the analysis are recorded in Supplementary Figure 2 and Supplementary Table 3.

Supplementary Figure 2. Estimation of protein turnover flux in the dataset. The average baseline turnover flux of proteins in control hearts of six strains of mice (A/J, BALB/cJ,

C57BL/6J, CE/J, DBA2/J, FVB/NJ) is estimated according to Hammond et al. 23. A: Scatter plot of protein turnover flux against protein turnover rate k (d⁻¹) (left) and protein abundance (right). Data point: protein species. Protein abundance span a larger dynamic range than turnover and exerts a larger effect on the overall protein turnover flux in the heart. B: Scatter plot of baseline turnover flux against log₂ ratio of turnover in hypertrophic vs. normal mouse heart (Spearman's $\rho = -0.19$). C: List of 20 myocardial proteins with highest flux in the dataset, after removing likely blood protein contaminants.

We agree with the Reviewer that flux analysis might present a valuable avenue in future investigations, but refrain from drawing further conclusions here as our primary goal is to compare the three omics parameters in detecting disease proteins. Flux measurements may not be directly compared to protein abundance in correlational analyses because the two parameters being compared (e.g., flux vs. abundance, with the former also being a function of abundance) would no longer be independent. Accurate comparison of flux quantity between normal and hypertrophic heart would likely also require absolute normalization with respect to heart weight.

Comment 9: Lastly, I do not believe that enough is made of the remarkable strain differences. The B6 mice show about a 4% HW:BW, the DBA 6.5%. But, what matters are the values at the start and end of the experiment -the changes in HW:BW are the important parameter - 109% in B6, 128% in DBA - are these changes significantly different. The DBA proteins should thus show greater labelling due to hypertrophy.

Response: We appreciate the Reviewer's comment. We now touch upon this important point raised by the Reviewer in the Discussion section:

Page 11 lines 31-33 *"Our data document a significant range of HW/BW ratios in various genetic strains, which may prompt future analysis on how these models may be interpreted in cardiac studies."*

The HW/BW data are strongly correlated with echocardiographic measurements in an independent study, thus externally validating the measurements (Supplementary Figure 7A; (Wang et al. 2016)). Please also see our response to Reviewer 1 Comments 5, 12, 14 for our rationale of adopting HW/BW ratios as a metric for hypertrophy in the study.

Comment 10: Simplify the arguments and assert the hypotheses more clearly. There are a lot of different analyses in here, but it is not that easy to work out their relatedness. It might be prudent to ask if some part of this work could be published elsewhere. What are the most important Q: transcript/ks relationships, pool size/flux balance? Growth corrected true increase in synthesis.

Response: We thank the Reviewer for this comment. We have made revisions to the introduction and conclusion sections to frame the primary findings, and now provide a revised abstract. The major hypotheses are now summarized in the manuscript:

Page 3 line 31 – Page 4 line 1 *"Comparing confident disease signatures that are consistently identified across six mouse strains, we observe that protein turnover, protein abundance, and transcript abundance nominate nonredundant candidate signatures of cardiac*

hypertrophy. Moreover, variations in protein turnover are independently associated with protein functional association and cardiac disease phenotype, suggesting a potential mechanism that uncouples various omics measurements.”

Comment 11: I am quite unpersuaded by the log₂ transform of kdeg values, which a) are limited in dynamic range, and the linear parameter. The compression of values might look nice, but the graphs are for the reader, and this is a distortion of the true behaviour.

Response: We thank the Reviewer for this comment. As stated in our response to Reviewer 1 Comment 2 above, we have now included linear representation in key figures (Figures 5C, 5D, Supplementary Figures 1E, 5A) for the readers. In addition, we would like to take this opportunity to address the Reviewer’s question on the suitability of casting turnover rate constant data in the logarithmic scale. The following considerations partially guided our choice of data treatment:

(i) As the Reviewer pointed out, protein turnover rates tend to follow log-normal distributions, resembling normality in the logarithmic space but not in the linear space. We believe this to be an intrinsic property of cellular protein turnover and one that has previously been observed in multiple studies by us and by others. Data points drawn from a log-normal distribution afford greater power for statistical inference when treated in the logarithmic space, due to the ability to make stronger assumptions in modeling data distribution over nonparametric methods (e.g., see Motulsky 2014 Intuitive Biostatistics 2 ed.).

(ii) Log-scale plots minimize high-leverage points that can disproportionately influence correlation and regression analyses. Data points are more homoscedastic with regard to errors in isotopomer measurements when cast in log scales, i.e., linear shifts in mass isotopomer ratios from mass spectrometry measurements correspond to proportionate deltas in log turnover but not linear turnover.

(iii) Gene expression data are conventionally presented in logarithmic scales (e.g., (Allison et al. 2006)). As one of our goals here is to compare multiple omics data types, casting the turnover data similarly in log scale and log ratios would be conducive to direct comparisons via log ratios and log-log correlations.

Comment 12: Sup T1 - add animal weights

Response: We have now added animal weight data to Supplemental Table 1.

Comment 13: Fig 1: simplify and clarify experimental design image - isoP starts at same time as D2O - this is actually important - see growth comments

Response: We have revised Figure 1A to clarify our experimental design. Please see also our response to Reviewer 1 Comment 3.

Comment 14: Figure 4A: Showing the HW:BW ratio (again, clarify this is %) is a little confusing, because, for example the DBA hearts are a higher % at the start - thus, the change is critical, rather than the actual proportion

Response: Following the Reviewer’s suggestions we have revised Figures 5C, 5D; and Supplementary Figures 6A, 6B, 7F, 7H to clarify the unit (mg heart weight per g body weight) commonly employed for HW/BW measurements. We have also included body weight data in Supplementary Table 2. Please also see response to Reviewer 1 Comment 5, 9, 12.

Comment 15: Figure 4C/D - there are six strains, and one measure, so why are there six bold points and six faint points - not explained in legend, and not easy to understand. The dark symbols actually cluster tightly with little evidence of a trend, so the faint symbols are important to the conclusion implied by the correlation diagrams. Again, I would prefer to see the log scaling removed - the dynamic range of kdeg values is small (by definition, since the labelling window is short)

Response: We thank the Reviewer for pointing out this potential misunderstanding. The gene correlation network analysis takes in HW/BW data from normal and hypertrophy hearts in each strain, equaling twelve data points per protein. The faint data point symbols are unintentional, and are the results of overlaps between the data points and the translucent areas of the confidence intervals of the trend lines. Following the Reviewer's suggestion, we have now recreated the figures and moved the data points to the front layer to avoid any potential misunderstanding.

Secondly, prompted by the Reviewer's suggestions we have also reanalyzed the data in linear scale for the Reviewer's consideration (Figures 5C, 5D). We observed the same trends in the differential regulation of contractile vs. metabolic proteins as before, although the randomly assigned color codes for the module have changed. The logarithmic scale figures are moved to Supplementary Figures 6A, 6B. Please also see discussion on linear vs. logarithmic scales in our response to Reviewer 1 Comment 2.

Comment 16: All figures that plot kdeg (is it really kdeg, if there is growth?) should indicate the dimensions (d). Cell biologists would probably interpret those numbers as h!

Response: We thank the Reviewer for the comment. We have now included dimensions (d^{-1}) in the axis labels of figures throughout the manuscript, including in Figures 5C, 5D, 6C; Supplementary Figures 5A, 6A, 6B, 7B, 7D.

Comment 17: Supp F1, is spectral counting as good as is needed for this analysis? - please reconsider

Response: We have now included a more detailed figure panel showing distribution of MS1 LFQ intensity and spectral counts of each sample in Supplementary Figure 1. Please also see our response to Reviewer 1 Comments 6 and 19 for the new analysis performed.

Comment 18: Supp F2, data would be better if we have absolute rates and errors on kdeg measure..

Response: We appreciate the Reviewer's comment. Following the suggestion we have now updated Supplementary Figure 5 (previously Supplementary Figure 2) to include absolute turnover rates in the revised manuscript. In addition, we have revised Supplementary Table 2 to include the basal absolute turnover rates and errors for the 273 consistently discovered candidate disease signatures.

Comment 19: Explain how the LFQ data were obtained - which samples? which time point in labelling? separate experiment at t=0?

Response: We thank the Reviewer for this comment. Briefly, protein MS1 LFQ may be measured as the sum of total area of each integrated isotopomer, using the same data, time points, and identical integration process as needed for turnover calculation. We now include additional description in the Methods section:

Page 15 lines 21-27 *“The areas-under-curve of up to the first six peptide isotopomers for each qualifying peptide at each time point are integrated over retention time space for abundance and turnover calculation. To calculate abundance, the summed areas of all isotopomers within each peptide envelope were normalized against total spectral intensity, then normalized against the number of possible tryptic peptides (6–30 amino acids) from in silico digestion of the protein. Abundance changes are presented as log₂ ratios of abundance at each hypertrophy time points over control day 0 samples.”*

Please also see response to Reviewer 1 Comments 6 and 17.

Manuscript

Comment 20: Make clear when the ms is talking about turnover (actually controlled by kdeg) rather than synthesis
Clarify the compensation data correction of ks for growth

Response: We thank the Reviewer for this suggestion and have now clarified our definition of the turnover rate constant (*k*) throughout the manuscript.

Page 4 lines 11-14 *“We developed a computational software, ProTurn, which integrates mass isotopomer signals of all peptides at multiple time points, and utilizes a custom kinetic model to calculate protein turnover rates (*k*), defined here as the rate (per day) at which a protein pool is proportionally replaced by a combination of synthesis and degradation processes”*

Comment 21: The animals were taken at the same time, and kept on a 12h light/dark cycle - but we are not told where in the activity/dark cycle the animals were - please explain

Response: Animal treatments including deuterium oxide and isoproterenol administration began at 12:00 noon of the corresponding day for all replicate groups, which is at 50% of the daily light/sleep cycle of the animal facilities (lights on 6 am to 6 pm). This is now reflected in the Methods section of the revised manuscript:

Page 13 lines 15-17 *“Upon arrival, the animals were housed in a 12 h/12 h light-dark cycle (lights on from 6 a.m. to 6.p.m. local time) with controlled temperature, humidity, and free access to standard chow and water.”*

Comment 22: This reviewer had never heard of inter-vigintile range, and neither has Google! (it threw up one paper, plus translations!). Please explain this term, or correct to what was meant.

Response: We refer to vigintile following the definition of “any of the values in a series that divides the distribution of individuals in that series into twenty groups of equal frequency.” (<https://en.wiktionary.org/wiki/vigintile>). However, we realize the modified word intervigintile is not in common use. Following the Reviewer’s suggestions, we have now replaced the term with the synonymous and more common “5th to 95th percentile”.

We would like to thank the Reviewer again for a thorough evaluation of the manuscript. It is our view that the manuscript has greatly benefitted from the Reviewer’s expert recommendations.

Reviewer #2 (Remarks to the Author):

Comment 1: Heart remodeling is an important physiological and pathological process whose regulation is still poorly understood- especially unclear is how the temporal dynamics of thousands of transcripts and proteins in many cell types (only ~50% of the heart cells consists of cardiomyocytes) is coordinated.

Lau et al use a combination of transcriptomic and cutting-edge proteomic methods and data mining to analyze the changes in transcripts and protein levels during isoprenaline-induced hypertrophic remodeling, one of several commonly used hypertrophy models.

The comprehensive use of six established mouse genetic lines and their comparative analysis is a particular strength of this work.

Response: We sincerely thank the Reviewer for a careful reading and critique of the manuscript. We especially appreciate the Reviewer's appraisal on the value of comparative analysis across genetic strains.

Comment 2: However, it seems a bit overblown to emphasize a "hitherto unknown association between metabolic remodeling in hypertrophy and coordinated protein turnover in selected sub-proteomes". Recent work e.g. on the coordinated turnover of myofibrillar proteins, metabolic enzymes and transcription factors by the muscle-specific TRIM ubiquitin E3 ligases MURF1/2/3 showed clearly that groups of cardiac protein turnover regulators target selected sub-proteomes- it would be useful to refer to some of these data and highlight changes of these established turnover regulators briefly, even if only in the supplements. Nevertheless, the comprehensive, not target-gene focused approach here reveals a large haul of novel data that will be valuable for cardiac research. Overall, this is an interesting, rigorously performed analysis that yields large amounts of novel data and insight into cardiac remodeling.

Response: We thank the Reviewer for this comment and were remiss not to highlight recent advances coming from complementary perspectives from studies on ubiquitin ligases. As the Reviewer has astutely pointed out, they reveal important information on the underlying molecular processes that can uncouple protein abundance from protein turnover, and provide mechanistic underpinning to any large-scale observations from omics studies. We have now expanded the Discussion section to include as a potential mechanism underlying observed proteomic changes:

Page 11 lines 28-31 *"In the heart, mounting evidence on the coordinated turnover of myofibrillar proteins, metabolic enzymes and transcription factors by the muscle-specific TRIM ubiquitin E3 ligases MURF1/2/3 also suggest cardiac protein turnover regulators can target selected sub-proteomes and exert post-transcriptional regulatory effects on overall protein abundance (Kedar et al. 2004; Maejima et al. 2014; Baskin and Taegtmeyer 2011; Fielitz et al. 2007)."*

Comment 3: While the resulting data are surely interesting, they very difficult to penetrate. 70 candidate disease molecules were identified, many of which apparently previously not associated with remodeling-associated turnover changes, but these results are somewhat forlorn in the supplemental tables and their functional context must be pieced together.

Response: We thank the Reviewer for this comment. We have now expanded on our discussion of the functional interpretation of these changes. Specifically, new figure panels have been included to more clearly show remodeling-associated turnover changes in various Reactome-annotated cellular pathways relevant to cardiac functions (Figures 2B-2C in the revised manuscript). Several figure panels have been revised for accessibility (new inset of

Figure 4B; revised Figure 4C; inclusion of additional functional categories in Figure 4D; selected interactome examples in Figure 6C). We have revised Supplementary Table 2 to illustrate the changes in each candidate disease signature. We have also clarified descriptions of the data throughout the text, specifically by revising paragraphs 2 - 4 (page 5 - 6) of the Results section to emphasize biological contexts alongside quantitative values. Please see also our response to Reviewer 2 Comment 4 below.

Comment 4: While this paper will be of immediate interest to the proteomics community, in its current form the figures and tables are of considerable complexity to the point of being almost impenetrable. Quite generally, the paper would significantly benefit from a more accessible presentation of the results and the key findings.

For example: surprisingly, sarcomeric proteins, which are recognized to undergo major changes during remodeling due to altered isogene expression, differential splicing and altered degradation kinetics do not feature recognizably. While the discussion alludes to contractile and sarcolemmal proteins, these are missing as a recognizable category in most figures. For most cardiovascular scientists, these would clearly be benchmark changes to put the data concerning other cellular pathways as well as cell types into perspective. Would it be possible to add contractile and sarcolemmal proteins to Figure 3 D and Supplemental Figure S3?

Response: We thank the Reviewer for this important comment. We agree that the manuscript might be more readily understood by readers with proteomics backgrounds and we were remiss not to highlight various cardiac protein categories. In this revised submission we have attempted to make the data accessible to a wider audience. We now include additional figure presentations to reflect pathway-level alterations in the turnover of proteins under 12 Reactome pathways relevant to cardiac functions, including R-MMU-390522 (striated muscle contraction) as suggested by the Reviewer (Figure 2B in the revised manuscript). We have revised Figure 3D (Figure 4D in the revised manuscript's numbering) to include additional gene ontology categories that are over-represented, including proteins involved in muscle contraction as suggested by the Reviewer.

We regret if any remaining underexplored aspects may persist. We hope the Reviewer agrees that due to the inherent information-richness of large-scale datasets, not every aspect of the data may be highlighted in the manuscript. Here we have emphasized our goal to determine whether the integration of omics data types facilitates the discovery of proteins and pathways among consistently altered proteins across multiple replicates/genetic backgrounds.

Figure 2B: Distribution of log ratios (hypertrophy/normal) in protein turnover (red), protein abundance (blue), and RNA abundance (gray) for protein members (data points) in 12 Reactome pathways relevant to cardiac functions

Comment 5: It is not necessarily surprising, but unprecedented in the depth of the analysis presented here, to see the turnover of protein-protein interacting partners being coordinated. Could exemplary pairs for the main functional groups be highlighted.

Response: We thank the Reviewer for the comment. We have now highlighted examples of known protein-protein interactors in the Figure 6C as well as in the text in the Results section of the revised manuscript:

Page 10 lines 5-7 *“Known protein-protein interacting pairs show similar turnover, including PKA and its anchoring protein AKAP1; and protein phosphatase 2A and its regulatory subunit (Figure 6C).”*

Figure 6C: Selected examples of interacting proteins with similar turnover

We have highlighted additional interaction pairs in the revised manuscript in Supplementary Table 6. Full data may be found in our open dataset on ProteomeXchange (PX000561) and Sage Synapse (doi:10.7303/syn2289125).

Comment 6: Supplemental Table S2 Disease_Prots shows truncated protein names in column 2 (PN), and would significantly benefit from a much clearer layout, complete protein names, and more explanation. This is after all one of the key results, identifying the 70 novel targets of cardiac remodeling. The table entries are incomprehensible for anyone but a mass spectrometry expert- what is FALSE and TRUE and what does it tell the reader about the data?

Response: We apologize for the suboptimal formatting on the PDF copy of the Supplementary Tables. We would like to assure the Reviewer that the full Supplementary Tables have also been made available in Excel spreadsheet format with all information and column labels intact. In this resubmission we have uploaded both the Microsoft Excel spreadsheet as well as a revised PDF copy with widened column lengths.

In addition, following the Reviewer's comment we have also appended additional information to the legends tab of the Supplementary Tables, and reproduced the legends in the first row of each spreadsheet for ease of navigation. We have replaced the binary flags denoting consistent up- or down- regulations in transcript, protein, and turnover experiments with more readable arrows. We have included additional legends and indication to highlight the 70 disease signatures nominated via turnover alone.

Comment 7: Columns “Group” and “Rep” are truncated in table Supplemental Table S4. What are the red highlights in this table?

Response: We apologize again for the suboptimal formatting on the PDF version of the Supplementary Tables. In the resubmission we have uploaded both the Microsoft Excel spreadsheet and an updated PDF with widened column lengths. Please also see response to Reviewer 2 Comment 6 above for additional changes to the Supplementary Table.

Comment 8: Line numbers in table descriptions in the supplements do not match up with the table referred to.

Response: We have revised the description of the Supplementary Tables in the revised manuscript. The order for Supplementary Tables 3 and 4 has been corrected. In addition, lines are merged to avoid confusion in line numbers and empty lines are no longer numbered. We apologize if any formatting mistakes persist in the revised submission.

Minor comments

Comment 9: Some smaller grammatical mistakes (e.g. ubiquitin in Fig 3D) and labels on subfigures missing: S3, D missing.

Response: These typographical errors have now been emended in the revised manuscript (Figure 4D, "ubiquitin"; Supplementary Figure 7H in the numbering scheme of the revised manuscript).

We would like to thank the Reviewer again for a thorough evaluation of our manuscript. We are in debt to the Reviewer's helpful suggestions and comments.

Reviewer #3 (Remarks to the Author):

Comment 1: The manuscript from Lau and co-workers used an isoproterenol-induced hypertrophy model to investigate the overlap of mRNA levels, protein abundance, and protein turnover. For the analysis of protein turnover rates the authors used deuterium levels and a previously developed model to calculate absolute turnover rates. By overlaying three independent datasets the authors potentially identified 70 proteins with a disease related function. Overall the systematic analysis of protein turnover rates and protein abundance is an important new field and opens a new direction how proteins are regulated during any perturbation. However, the manuscript is very descriptive and absolute no functional analysis were performed.

Response: We thank the Reviewer for a thorough evaluation of our manuscript and for the many helpful suggestions. Please see below our itemized responses.

Major points:

Comment 2: The authors should explain the workflow proteins abundance were measured in the results part. It is not clear why the authors identified more than 8000 proteins and quantified only 3200 proteins. What happens with the other 5000 proteins.

Response: We thank the Reviewer for pointing out this potential misunderstanding. The difference is due to sequential filtering steps taken to ensure data quality in proteins considered confidently identified vs. confidently quantified. We employed stringent criteria for protein quantification as described the open dataset descriptor (Lau et al., 2016), which we summarize below.

Protein turnover quantification required higher evidence burden than protein abundance identification, as it requires (i) explicit identification of the protein and its proteome-unique peptides in at least four data points; (ii) a peptide isotopomer time-series must fit to the kinetic model with a goodness-of-fit (R^2) of ≥ 0.8 or standard error of estimate (s.e.) ≤ 0.05 to be

considered to pass the stringency filter for turnover quantification. These criteria were optimized by targeting greater numbers of quantifiable proteins while minimizing intra-protein variability, and resulted in concordant turnover rate constant values among unique peptides of a protein (Lau et al., 2016). Hence a number of proteins were confidently identified but did not lend themselves to high-quality turnover quantification. We have previously noted that these proteins may be candidates for more targeted analysis, e.g., they may be very-long-half-life proteins that do not accumulate isotope labels readily. Following the Reviewer's comments we now expanded the Methods section to describe the workflow by which proteins are quantified:

Page 15 line 31 - Page 16 line 3 *"Only a subset of peptide isotopomer time-series with goodness-of-fit ≥ 0.8 or standard error of estimate ≤ 0.05 were accepted for turnover calculation (Lau et al., 2016). Turnover rate was reported as the median and median absolute deviation of the best-fitted turnover rate constant of each accepted constituent peptide that is unique in the proteome."*

Comment 3: In addition, the comparison to the mRNA dataset revealed only 1900 overlapping proteins. One should expect that each protein is at least expressed and present in the mRNA database. The authors stated that the overlap of protein turnover and protein abundance is poorly correlated but did not explain/discuss this inconsistency.

Response: We thank the Reviewer for pointing out this potential misunderstanding. The Reviewer is correct to point out that the mRNA datasets cover the quantified proteins. The number 1901 refers to the number of transcript/proteins with observations (confident quantification in turnover, protein, and RNA) that may be compared in all strains, which we used as the basis of the inter-strain comparison in the subsequent analysis. Please also see our response to Reviewer 3 Comment 2 above for filtering criteria. Because of our data filtering criteria, some proteins only have turnover rates confidently quantified ($R^2 \geq 0.9$) in a subset of mouse strains. This has now been made more clear in the manuscript:

Page 4 lines 30-32 *"A subset of 1,901 proteins that are commonly shared in all six strains has been selected for analysis; as these 1,901 proteins possessed quantified transcript abundance, protein abundance, and protein turnover data across the six strains."*

Comment 4: The authors selected six mouse strains and it seems that each strain were analyzed only once without any biological replication. It would be nice to have at least for some mouse strains a biological triplicate analysis to judge the significance of the dataset.

Response: We thank the Reviewer for pointing out this potential misunderstanding. In our experimental design up to 7 replicate groups of animals provided independent measurements which independently contributed to parameter estimation in the best-fitted values of k , and their residuals from the best-fitted curve independently contributed to the estimation of variance. This is a commonly accepted method for time course and kinetics data and has been described by us and by others extensively in previous publications (Claydon et al., 2012; Hammond et al., 2016; Kim et al., 2012; Price et al., 2012; Wang et al., 2014). We have previously demonstrated that the variance of k can be estimated both using a bootstrapping technique that resamples measured isotopomer distribution data, or an analytical method to solve for dk/dA as employed in the current dataset. Both methods allow statistical inference to judge significance of changes in individual strains. For details, see also "Strategies for data quality assurance" section in (Lau et al., 2016). Please also see our response to Reviewer 3 Comments 6 and 20, as well as Figures R2 and R3, below.

We would like to note that although in theory data from one time point is sufficient for kinetic modeling, and that time point may be repeatedly measured in biological replicates, the spread of replicates across multiple time points along the kinetic curve is preferred for protein turnover studies (see also (Claydon et al., 2012; Hammond et al., 2016)). Following the Reviewer's comments, we have also modified the Methods section to clarify the methodology:

Page 15 lines 27-31 *"To calculate turnover rates, the fractional abundance of the first mass isotopomer from each integrated time point (A_0) was modeled with a compound kinetic curve using the Nelder-Mead algorithm. The A_0 values measured by the mass spectrometer from each of seven time point served as independent biological replicates to allow estimation of the best-fitted values as well as variance of the turnover rate constant (k). Only a subset of peptide isotopomer time-series with goodness-of-fit ≥ 0.8 or standard error of estimate ≤ 0.05 were accepted for turnover calculation."*

Comment 5: The cut-off seems to be arbitrary. Why have the authors selected a cut-off of 1.4 for increased turnover and decreased turnover 1.2.

Response: We thank the Reviewer for pointing out this potential misunderstanding. Criteria for differential turnover was based on their rank percentile and Z scores within each dataset. No magnitude-based cutoff was applied. The observation that significantly increased turnover had the values of 1.4 and above was simply reported to describe the overall more prevalent increase than decrease in turnover among the sampled proteins.

Following the Reviewer's comments we have revised the paragraphs in questions for clarity:

Page 5 lines 6-8 *"We define a putative consistent disease signature here as a gene/protein exhibiting changes in 80th or above percentile in ranks in four or more genetic backgrounds, followed by a combined test for significance"*.

Comment 6: In the case the authors performed biological replicates one should perform a FDR analysis based permutation based statistics. The chapter on page 3 (unfortunately no page numbers are present) line 23-36 is difficult to understand and should be rephrased. For example "These proteins are..." which one? And "They preferentially function..."

Response: We thank the Reviewer for this comment. As discussed above in our response to Reviewer 3 Comment 4 above, the turnover rate data within each strain and condition were derived from four to seven biological replicate groups of animals, which provided independent observations in isotopomer incorporation along the kinetic curve (see Figure R2 below). As discussed above, our group have previously demonstrated that fitting error can be estimated both using a bootstrapping technique that resamples measured isotopomer distribution data, as well as an analytical method that solves for the derivative of k with respect to fractional isotopomer abundance (Lam et al., 2014). The fitting data including best-fitted rate constants, fitting error, and goodness-of-fit are fully available for exploration and re-analysis in our open dataset, which is accessible on ProteomeXchange (PX000561) and Sage Synapse (doi:10.7303/syn2289125), and has been described in greater detail in the dataset descriptor (Lau et al., 2016). In the revised submission, we have included data and code availability statements and encourage the readers to access all 120,454 peptide time series that pass the filtering criteria along with the estimation of variance, which can be found on our open data repository submissions. Please also see responses to Reviewer 3 Comments 4 and 20.

R2

Protein: Q91VI7 (Ribonuclease inhibitor RNH1)
Peptide sequence: LLCEGLQDPQCR²⁺
Data points: 7
Best-fitted rate constant: 0.147 d⁻¹
Confidence interval: [0.132 - 0.165] d⁻¹
Goodness-of-fit R²: 0.987
Standard error: 0.049
N (deuterium-accessible labeling site): 27
k_p (total deuterium enrichment rate constant): 10 d⁻¹
a (Pre-enrichment fractional isotope abundance): 0.436
p_{ss} (plateau deuterium enrichment): 0.051
Kinetic function: $dA_0/dt = k \cdot (a \cdot (1 - p_{ss}) \cdot (1 - e^{-k_p t})^N - A_0)$
Optimization algorithm: Nelder-Mead

Figure R2. Estimation of variance in kinetic curve fitting. The kinetic time-series of an individual peptide (LLCEGLQDPQCR from RNH1) in C57BL/6J mouse is shown. Data points from seven independent replicate groups of mice collected at days 0, 1, 3, 5, 7, 10, and 14 are used to find best-fitted values of turnover rate constant k . Error of fitting is estimated from residuals to the curve. Adopted from our open dataset via figshare (<https://doi.org/10.6084/m9.figshare.c.2171334>)

Secondly, following the Reviewer's suggestions, we have rephrased the paragraph in question for clarity. It now reads:

Page 6 lines 8-22: *“Cellular pathways relevant to cardiac functions show diverse changes across omics data type (Figure 2B), whereas signatures nominated from each omics type are enriched in different functional categories. Turnover signatures are enriched with proteins located in the extracellular matrix (Enrichment (E): +6.6-fold; hypergeometric test with Benjamini-Hochberg adjusted P-value (P): 3.2e-3), sarcolemma (E: 7.5; P: 2.8e-3), cellular potassium ion homeostasis (E: 39.5; P: 2.4e-3), glycolysis (E: 8.3; P: 2.2e-2). Proteins with consistently decreased turnover are more likely to localize to the mitochondrial matrix (E: 4.9; P: 1.7e-3) and function in fatty acid beta-oxidation (E: 10.9; P: 3.9e-3). Signatures with increased protein abundance in hypertrophy are statistically more likely to function in muscle contraction (E: 16.2; P: 7.1e-5), whereas those with decreased abundance are enriched in peroxisomal membrane (E: 8.3; P: 8.7e-3). Transcript up-regulation is observed preferentially in extracellular space (E: 9.2; P: 6.1e-12), cell surface (E: 4.6; P: 3.1e-3), regulation of inflammatory response (E: 19.5; P: 1.5e-3), and proteolysis (E: 4.5, P: 8.6e-3). Downregulated transcripts are preferentially in mitochondrial proteins (E: 3.4; P: 2.8e-4), involved in fatty acid beta oxidation (E: 9.4; P: 1.0e-2). Hence, the data here suggest that disease signatures associated with different aspects of cardiac functions may be discovered from each omics data type, and are consistent with specific regulatory modalities in each cardiac subproteome.”*

Comment 7: The authors stated a “modest overlap” please quantify this. It might be useful to calculate the AUC (area under the curve) for some proteins with matching and non-matching protein abundance and turnover. Unfortunately, the first part is difficult to understand and the overlap of all proteins between different conditions is definitely very confusing for readers which are not in the proteomics field.

Response: We apologize for the potential ambiguity in this paragraph. The sentence in question referred to Figure 1 and were addressed in the subsequent paragraphs. Following the Reviewer's suggestions, we have now revised the paragraphs for clarity and included numerical values where applicable:

Page 4 line 32 – Page 5 line 2 *“As expected, we observed poor correlations between changes in protein turnover in hypertrophy with changes in protein abundance ($\rho = 0.11$) (Figure 1B), changes in transcript abundance ($\rho = 0.09$) (Figure 1C), or basal turnover flux (Supplementary Figure 2).”*

Page 5 lines 10-17 *“They include 49 genes/proteins with consistently increased protein turnover, 50 with decreased protein turnover; 61 with increased protein abundance, 55 with decreased protein abundance; 59 with increased transcript abundance, and 51 with decreased transcript abundance. We reason that if disease signatures in hypertrophy are primarily driven by transcript-level changes, similarity between reproducible disease signatures at transcript and protein levels would be expected. Instead, we observed low commonality in candidate disease proteins between the turnover disease signatures with protein abundance signatures (12%) or transcript abundance signatures (18%).”*

Comment 8: The authors identified 70 proteins with altered turnover. What is the proportion of enhanced and decreased turnover here? What is the significance of this candidates? The authors described that DHRS7C as a protein with decreased protein turnover but increased protein abundance. It would be nice to confirm this by an alternative approach and it be that the ubiquitination is also affected to stabilize the protein. Otherwise this part remains quite speculative.

Response: We thank the Reviewer for this comment. We have now stated that 37 out of 70 (53%) consistently identified candidates with altered turnover had increased turnover; whereas 33 out of 70 (47%) had decreased turnover:

Page 6 lines 24-26: *“Further analysis on the 70 proteins discovered via turnover alone (37 proteins with increased turnover and 33 with decreased turnover), suggests they also contain proteins that are functionally associated with the remaining 203 candidate disease proteins (out of 273).”*

We have now expanded our discussion on the biological interpretation of the results throughout the manuscript (e.g., in Figures 2C-D, 6C and accompanying texts). We agree with the Reviewer that the stated hypothesis on DHRS7C will be tested in future work. Our primary goal in this section is to illustrate the discovery of contra-directional changes in each of the three omics parameter. We have now revised the paragraph following the Reviewer's suggestion:

Page 5 line 31 – Page 6 line 3 *“Conversely, a considerable proportion (38%) of disease signatures show inconsistent or discordant directions in their changes of transcript abundance, protein abundance, or protein turnover in hypertrophy (Figure 2A). For instance, dehydrogenase/reductase SDR family member 7C (DHRS7C) shows consistently repressed turnover amid isoproterenol challenge in five out of six mouse strains (1.3–2.7-fold) whilst its protein abundance is increased (1.2–1.7-fold). DHRS7C is a cardiomyocyte-expressed ER dehydrogenase previously reported to be transcriptionally down-regulated in heart failure (Lu et al. 2012); we speculate that this transcriptional downregulation might be balanced by decreased protein degradation, leading to protein accumulation.”*

Finally, although our goal here is not to focus on mechanisms regulating the abundance and turnover of individual protein targets, we now provide additional data suggesting that the measured turnover rate changes are influenced in part by changes in protein degradation. In C57BL/6J mice administrated for 14 days with epoxomicin (0.5 mg/kg/d i.p.), a selective proteasome inhibitor, an overwhelming majority of proteins (83%) show a decrease in effective protein turnover rates (Supplementary Figure 4). This result is consistent with a post-translational origin in the regulation of protein turnover rates as presented in the manuscript.

Comment 9: The authors found increased mRNA levels of “inflammatory response”. This might reflect the presence of other cell types such as macrophages.

Response: We thank the Reviewer for a thought-stimulating comment. We agree that changes in individual cell types may underpin various observed disease phenotypes; such confounding factors would be endemic to all whole-heart models used in the literature. Interestingly, we and others have observed that the turnover rates of even the same protein may differ between the pools in different tissues and cell types (Hammond et al., 2016; Kim et al., 2012), and that turnover rates might be used to discern the tissue-of-origin of certain proteins (e.g., blood contaminants in the liver and erythrocyte contaminants in the plasma).

Although we fully agree that the acquisition of new omics data from individual cell types would be of great scientific interest, the primary goals of our study are to (i) compare the overlaps and non-overlaps of consistent candidates among three omics parameters across multiple genetic backgrounds, and to (ii) evaluate the application of protein turnover rate measurement as a means to discover disease proteins and pathways in a commonly employed and physiologically relevant disease model. We hope the Reviewer will agree that the collection of new data from different cell types will require substantial time and lie beyond the scope of the current experiments. Following the Reviewer’s comments we now discuss this as a potential future avenue in the Discussion section.

Comment 10: Overall the discussion fails to explain the reason for the divergent regulation of protein abundance and turnover.

Response: We thank the Reviewer for this comment. We have expanded the Discussion sections to discuss our views potential mechanisms on the divergent regulations of protein abundance vs. protein turnover.

Page 11 lines 24-31 *“Changes in protein abundance may be uncoupled from changes in protein turnover by mechanisms of protein degradation, which may be mediated by the 600-1,000 E3 ubiquitin ligases encoded in the human genome, e.g., when protein degradation is stimulated, a measurable increase in turnover may result in parallel to a decrease in overall protein abundance (Lam et al. 2014). In the heart, mounting evidence on the coordinated turnover of myofibrillar proteins, metabolic enzymes and transcription factors by the muscle-specific TRIM ubiquitin E3 ligases MURF1/2/3 also suggest cardiac protein turnover regulators can target selected sub-proteomes and exert post-transcriptional regulatory effects on overall protein abundance (Kedar et al. 2004; Maejima et al. 2014; Baskin and Taegtmeier 2011; Fielitz et al. 2007).”*

Comment 11: In addition, the combination of six mouse strains and the hypertrophy model seems to be not very helpful to identify new proteins regulated by hypertrophy. A more focused analysis of one strain (such as in the mRNA datasets) might be sufficient to describe the effect and reduces the complexity of the presented data.

Response: We thank the Reviewer for the important comment and agree that considering one strain first would help dissect the complexity of the dataset. A more focused data analysis on the C57BL/6J strain of mice revealed a number of insights, including: (i) cardiac proteins are renewed at a median rate of ~11% per day; (ii) cardiac hypertrophy leads to widespread alterations in protein dynamics, with significant increases in the turnover of proteins including FHL1 (+4.8-fold), FBN1 (+3.8-fold), and ANXA2 (2.1-fold); (iii) protein pathways including vasculature development, extracellular matrix, and membrane-bound vesicles show differential turnover in cardiac hypertrophy; and (iv) changes in protein turnover and protein abundance are poorly correlated.

We believe these are important findings; however, when we analyzed datasets from all six strains, we gained additional insights that were missing from analyzing a single strain alone. Firstly, data from the six strains allowed us to identify disease signatures that are commonly observed across diverse genetic backgrounds, which we believe to be more likely to be biologically relevant and broadly applicable, i.e., less likely to be dependent on a particular model. Secondly, the data allowed us to test whether these presumably more reliable disease signatures have more concordant changes among protein turnover, protein abundance, and transcript abundance. We found that the reproducible signatures nominated from each of the three omics parameters overlap poorly, hence providing evidence that they reflect different aspects of cardiac pathology. Thirdly, data from the six strains combined enabled gene-trait correlation analysis to be performed through the turnover rates differences among the strains. The inter-strain differences as shown in Figures 5C-D were relatively minor compared to the inter-protein differences as shown in Figure 4D, as most proteins show highly similar turnover across the dynamic range of measurable turnover rates, but the differences were sufficient to show correlation with the differential responses to isoproterenol in these strains.

The observations above are made possible by the analysis of multiple mouse strains. Because the mouse strain data are additive, we anticipate that they will gain in value in future studies to allow association studies using “turnover networks” in greater numbers of mouse strains. To help present the complex data we have added additional figure panels for different audience in this revised submission, including Figures 2B-C, 4B, and 6C.

Minor points:

Comment 12: Usually, histones are very stable proteins. Which half-life or turnover rate did were measured for HIST1H4A? Have the authors used only unique peptides for this histone? It might be that a prolonged labelling time (more than two weeks) will give much sharper results.

Response: We thank the Reviewer for an interesting comment. Only unique peptides (peptides found in no other protein entries in the UniProt complete proteome) are employed for the turnover analysis of all proteins, in order to avoid confounding of measured isotope incorporation rates from peptides shared multiple proteins. We have now clarified this technical point in the Methods section to emphasize that only proteome-unique peptides are used for turnover calculation

Page 16 lines 1-4 *“Protein-level turnover rate was reported as the median and median absolute deviation of the best-fitted turnover rate constant of each accepted constituent peptide that is unique in the proteome.”*

The Reviewer is correct to point out that histones tend to be stable proteins in the heart. Indeed in our dataset HIST1H4A has a half-life of ~55 to 88 days in the mouse heart; it corresponds to the top 1 percentile of most stable proteins in our dataset. Following two weeks of labeling the protein has reached ~15% of its half-life, which is within the limit of detection in our experimental design. Longer labeling will allow more power to discern finer differences in

very-slow-turnover proteins. Our group has previously subjected laboratory mice to deuterium labeling for up to 90 days at 5% deuterium oxide (Kim et al., 2012). In balancing time and material costs with the experimental design to compare turnover in multiple strains and conditions, we have elected to label for two weeks, based on previous experience that the median half-life of proteins in the mouse heart is approximately 7 days.

Prompted by the Reviewer's comments, we now also include the basal turnover rates for the 273 consistently discovered candidate disease signatures in Supplementary Table 2. The full data of all 120,454 confidently quantified peptide ion kinetic curves in our open dataset are publicly available on ProteomeXchange (PX000561) and Sage Synapse (doi:10.7303/syn2289125).

Comment 13: What is the mRNA level of DHRS7C in the mRNA dataset? It might be helpful to integrate it in Fig. 1E.

Response: We thank the Reviewer on this comment. Following the Reviewer's suggestions, we have now incorporated the changes of DHRS7C into Figure 2A for comparison. The DHRS7C expression level data also remain represented in Figure 3E.

Comment 14: "Rank aggregation algorithm" should be explained shortly in order to allow the reader to follow the subsequent section.

Response: Following the Reviewer's comments, we have now expanded on the description of methods for the meta-analysis in Figure 2 (Figure 3 in the revised manuscript). The reader is also referred to the reference (Ref. 42) for details.

Page 7 lines 18-20 *"The gene lists were subsequently aggregated by considering their overlaps in top-ranked genes against permutations, in order to evaluate the rank similarities in changes in transcript abundance, protein abundance, and protein turnover."*

Page 17 lines 13-15 *"The similarity of gene lists in the meta-analysis of multi-omics datasets was compared using the OrderedList package (v.2.4) in R/Bioconductor."*

Comment 15: Figure 3B and line 27, page 6: "We observed that Type I and Type II proteins differed from Type III proteins in that they exhibited greater expression changes during hypertrophy".

It is hard to observe this statement in Figure 3B. Another representation might be more appropriate. Is the difference significant?

Response: We appreciate this comment. Prompted by the Reviewer's suggestions, we now include an additional inset panel in Figure 3B (Figure 4B in the revised manuscript - the numbering has shifted due to the revision to panels in Figure 1 based on the Reviewers' suggestions) to compare the distributions of absolute log ratios of protein expression in day 1 and day 14 of hypertrophy among the three protein clusters. We have also included *P* values from nonparametric Mann-Whitney U tests to show the statistical significant differences between groups.

Comment 16: The permutation tests for interactome analysis are not described. Figure 5E,F are not mentioned in the text. Page 8, line 35: Figure5E should be Figure 5G

Response: We have now amended the reference in question to read Figure 6G in the revised manuscript (page 10 line 19). All figures including Figure 6E-F in the revised manuscript (Figure 5E-F in the previous submission) are now referred to in main text (page 10, lines 12-15). Following the Reviewer's suggestions, we have also expanded the description of the permutation tests in the Methods section, which now reads:

Page 16 lines 17-21 *“Permutation of interaction networks was done via label exchange by randomly sampling without replacement each of all curated interacting partners with quantified turnover rates then calculating the absolute distance in \log_2 turnover rates between each pair. Average numbers of interactions among proteins in each category (slow, medium, and fast turnover) after 10 million repeated permutations were reported.”*

Comment 17: In general, the axis titles are often not very clear, i.e. it should always be indicated on the axis which ratio or which turnover rate is shown e.g. hypertrophic/normal.

Response: We thank the Reviewer for pointing out this oversight. In the revised manuscript we have now more clearly labeled the figure axes to specify comparisons (e.g., hypertrophy/normal), including in Figures 2B, 3E, 4B, 4C; Supplementary Figure 5B.

Comment 18: Figure 2D: Do not demonstrate the protein abundances high rank similarities between themselves?

Response: We thank the Reviewer for an interesting observation. We believe that whilst segments of experiments on protein abundance show significant rank similarities among themselves, but similarities among protein turnover are more subtle. In our experience similarities among turnover datasets may be better observed seen when an arbitrary cutoff of $P < 0.1$ as opposed to $P < 0.05$ is employed, but we have elected a more conventional cutoff here. The fundamental similarities among turnover datasets can also be observed in the biweight midcorrelation plot in Figure 3C.

Comment 19: Suppl. Fig.2A: It gets not clear if for left and right plot the same amount of proteins was used. Relative Frequency instead of absolute frequency should be used.

Response: We thank the Reviewer for the suggestion. The left and right plots do indeed share identical axes and scales. We have now amended the figure panel of Supplementary Figure 5A (numbering following the revised manuscript) for clarity. Following the Reviewer's suggestion, we have also rescaled the axes of each pair of panels based on the counts (relative frequency) (see below).

Comment 20: Suppl. Fig. 2B: Only one replicate was measured per mouse strain. Shouldn't there be more replicates per strain to draw conclusions about strain specific turnover. Couldn't be the ratios just be observed by chance?

Response: We thank the Reviewer for bringing up a potential point of misunderstanding in the protein turnover analysis. We have implemented multiple measures to ensure data rigor. We would like to suggest to the Reviewer the following points for consideration.

- Each peptide isotopomer series per mouse strain constitute measurements at 4 to 7 time points, which originate from seven independent replicate groups of animals. The data points at each time point contribute independently to the multivariate optimization procedure to estimate the turnover rate constant k and its various. They should therefore be construed as true biological replicate. This experimental design is typical among protein turnover methods reports in the literature. We rigorously retain only well-fitted kinetic curves using goodness-of-fit and standard error metrics. We have previously described methods to estimate errors and confidence intervals from the kinetic curves using both the analytical solution to dk/dA as well as more conservative Monte Carlo bootstrapping methods (Kim et al., 2012; Lau et al., 2016).
- The consistency of measurements is further corroborated by replicate observations from multiple sister tryptic peptides from each protein, which constitute independent measurements of isotope enrichment from different ion species in the mass spectrometer. We have previously demonstrated the reproducibility of the measured turnover rates of peptides in the same protein in each strain, and calculated the median geometric coefficient of variance among peptides from each protein in the dataset (Lau et al., 2016). The standard errors (s.e.) of fitting of each peptide time-series are available on our open dataset on Synapse for the interested readers (doi:10.7303/syn2289125).
- Data from individual strains show that for the majority of proteins there is high consistency of measured turnover rates when considered across the entire dynamic range of the data (Figure 4D).
- We have now analyzed the mass spectrometry data from an independent series of animals in C57BL/6J. A summary of the data (median k for each Uniprot accession) has been uploaded to Synapse (accession syn8545245). Principal component analysis and t-distributed stochastic neighbor embedding plots below show that the two independent sets of data from C57BL6/J clearly cluster with one another, despite being acquired over a year apart (see Figure R3 below).

R3

Figure R3. Data distributions by sample. Plots from two dimensionality reduction analyses on the measured turnover rates of 734 common proteins across 13 samples are shown. Principal component analysis (PCA) (left) and t-distributed stochastic neighbor embedding (t-SNE) corroborate that samples from normal (nor; orange) and hypertrophy (hyp; blue) may be separated from each other. Furthermore, an independent repeat set of experiments on C57BL/6J normal mice (c57_nor and c57_nor_rep) shows co-clustering within identical mouse strain. Note that t-SNE dimensions are nonlinear and represent probability distribution rather than Euclidean distances between samples.

In summary, we employed stringent filtering criteria which we have previously described and validated against independent protein turnover datasets (Lam et al., 2014). Nevertheless we appreciate the reviewer bringing up this potential source of misunderstanding and were remiss in not explaining it in detail. We have now expanded on the Methods section to clarify our data quality control methodology (see also our responses to Reviewer 3 Comments 4 and 6 above).

Comment 21: Suppl. Fig. 2C: Figure is too small and individual proteins are not readable.

Response: This panel has now been enlarged to increase legibility.

Comment 22: Suppl. Fig.3: "D" is missing in the labelling.

Response: This has now been corrected.

We would like to thank the Reviewer again for a thorough evaluation of our manuscript. We feel that the manuscript has benefitted greatly from the Reviewer's inputs and suggestions.

References

Allison, D.B., Cui, X., Page, G.P. and Sabripour, M. (2006). Microarray data analysis: from disarray to consolidation and consensus. *Nature Reviews. Genetics* 7(1), pp. 55–65.

Claydon, A.J., Thom, M.D., Hurst, J.L., and Beynon, R.J. (2012). Protein turnover: measurement of proteome dynamics by whole animal metabolic labelling with stable isotope

labelled amino acids. *Proteomics* 12, 1194–1206.

Hammond, D.E., Claydon, A.J., Simpson, D.M., Edward, D., Stockley, P., Hurst, J.L., and Beynon, R.J. (2016). Proteome dynamics: tissue variation in the kinetics of proteostasis in intact animals. *Mol Cell Proteomics* 15, 1204–1219.

Heineke, J., Auger-Messier, M., Correll, R.N., Xu, J., Benard, M.J., Yuan, W., Drexler, H., Parise, L.V., and Molkentin, J.D. (2010). CIB1 is a regulator of pathological cardiac hypertrophy. *Nat Med* 16, 872–879.

Hill, J.A., Rothermel, B., Yoo, K.-D., Cabuay, B., Demetroulis, E., Weiss, R.M., Kutschke, W., Bassel-Duby, R., and Williams, R.S. (2002). Targeted inhibition of calcineurin in pressure-overload cardiac hypertrophy. Preservation of systolic function. *J Biol Chem* 277, 10251–10255.

Kim, M.-S., Pinto, S.M., Getnet, D., Nirujogi, R.S., Manda, S.S., Chaerkady, R., Madugundu, A.K., Kelkar, D.S., Isserlin, R., Jain, S., et al. (2014). A draft map of the human proteome. *Nature* 509, 575–581.

Kim, T.-Y., Wang, D., Kim, A.K., Lau, E., Lin, A.J., Liem, D.A., Zhang, J., Zong, N.C., Lam, M.P.Y., and Ping, P. (2012). Metabolic labeling reveals proteome dynamics of mouse mitochondria. *Mol Cell Proteomics* 11, 1586–1594.

Lam, M.P.Y., Wang, D., Lau, E., Liem, D.A., Kim, A.K., Ng, D.C.M., Liang, X., Bleakley, B.J., Liu, C., Tabaraki, J.D., et al. (2014). Protein kinetic signatures of the remodeling heart following isoproterenol stimulation. *J Clin Invest* 124, 1734–1744.

Lau, E., Cao, Q., Ng, D.C.M., Bleakley, B.J., Dincer, T.U., Bot, B.M., Wang, D., Liem, D.A., Lam, M.P.Y., Ge, J., et al. (2016). A large dataset of protein dynamics in the mammalian heart proteome. *Sci Data* 3, 160015.

Price, J.C., Holmes, W.E., Li, K.W., Floreani, N.A., Neese, R.A., Turner, S.M., and Hellerstein, M.K. (2012). Measurement of human plasma proteome dynamics with (2)H(2)O and liquid chromatography tandem mass spectrometry. *Anal Biochem* 420, 73–83.

Rau, C.D., Wang, J., Avetisyan, R., Romay, M.C., Martin, L., Ren, S., Wang, Y., and Lusis, A.J. (2015). Mapping genetic contributions to cardiac pathology induced by Beta-adrenergic stimulation in mice. *Circ Cardiovasc Genet* 8, 40–49.

Rau, C.D., Romay, M.C., Tuteryan, M., Wang, J.J.-C., Santolini, M., Ren, S., Karma, A., Weiss, J.N., Wang, Y., and Lusis, A.J. (2017). Systems Genetics Approach Identifies Gene Pathways and Adamts2 as Drivers of Isoproterenol-Induced Cardiac Hypertrophy and Cardiomyopathy in Mice. *Cell Syst* 4, 121–128.e4.

Rockman, H.A., Ross, R.S., Harris, A.N., Knowlton, K.U., Steinhilber, M.E., Field, L.J., Ross, J., and Chien, K.R. (1991). Segregation of atrial-specific and inducible expression of an atrial natriuretic factor transgene in an in vivo murine model of cardiac hypertrophy. *Proc Natl Acad Sci U S A* 88, 8277–8281.

Wang, D., Liem, D.A., Lau, E., Ng, D.C.M., Bleakley, B.J., Cadeiras, M., Deng, M.C., Lam, M.P.Y., and Ping, P. (2014). Characterization of human plasma proteome dynamics using deuterium oxide. *Proteomics Clin Appl* 8, 610–619.

Wang J.J., Rau C., Avetisyan R., Ren S., Romay M.C., Stolin G., Gong K.W., Wang Y., Lusic A.J. (2016). Genetic Dissection of Cardiac Remodeling in an Isoproterenol-Induced Heart Failure Mouse Model. *PLoS Genet.* 6, e1006038.

Wilhelm, M., Schlegl, J., Hahne, H., Gholami, A.M., Lieberenz, M., Savitski, M.M., Ziegler, E., Butzmann, L., Gessulat, S., Marx, H., et al. (2014). Mass-spectrometry-based draft of the human proteome. *Nature* 509, 582–587.

Reviewers' comments:

Reviewer #1 (Remarks to the Author):

Thank you for taking the time to address my concerns, even if the response to reviewers is now substantially larger than the original manuscript! Overall, I believe the authors have moved a good way to answering my concerns.

I have one residual concern - whatever the linear model might say, the growth curves (Figure R1) show significant growth for, minimally, A/J, B6, and DBA mice. I am also concerned about the variance in recorded weights. Are these weights from the study animals used for the turnover study - I don't really see how, since the numbers of animals would then be reducing for each time point, as most individuals are taken? Why no 10d and 14d d for FVB?

The variance, and the trends are remarkably noisy, and I would want to see these checked, perhaps on an individual basis. - we do not see such variance in our experience - is there a problem with husbandry? If there was any issue of food availability, then the study would be compromised. The text about animal maintenance should have ensured this variance was more controlled so it remains a curious observation.

However, the biggest issue is the growth of the animals, not the variance in body weight.

Although the sampling regiment is robust to body weight bias, there can be little doubt that in some of these strains, the total pool was expanding - in A/Js for example, by 20% over the experiments duration. Thus the incorporation reflects two terms - turnover and growth. The linear model was not set up to reveal this. I do think this need to be addressed. How much of the variation between strains related to their growth differences? On top of this, the isoP also expands the cardiac pool.

I don't think this should delay publication, but I would prefer a clearer discussion of this issue, and clarification that this is a measurement of one parameter that conceals two processes. To illustrate, Supp Fig 6A refers to 'turnover clusters' ...to my mind, these are 'synthesis clusters'?

Label incorporation (synthesis) is not the same as turnover (the dual processes of synthesis and degradation). Similarly, growth will increase apparent synthesis rates in the absence of degradation. I do hope the authors will be clear in the text about what they are measuring, and how it relates to their findings.

Thus statements in the text:

"utilizes a custom kinetic model to calculate protein turnover rates (k), defined here as the rate (per day) at which a protein pool is proportionally replaced by a combination of synthesis and degradation processes" is incorrect, if the system is expanding (I believe it is, from the growth data).

Epoxomicin experiment: inhibition of degradation is referred to as decreased turnover (the degradative component) whereas in Supp F2A, turnover refers to label incorporation. Also, if k_d is reduced, why do most proteins decrease in abundance? Was there a full, vehicle-control for this study?

It's just all a bit confusing - there is no doubt that important behaviours are being seen, but at the same time, it is actually difficult to tease apart mechanistic insights.

Figure 4: how much of the variance is explained by PC1 and PC2?

Reviewer #3 (Remarks to the Author):

The authors have addressed all questions and concerns of the reviewer.

Responses to Reviewers

--

Reviewer #1 (Remarks to the Author):

Reviewer 1 Comment #1: *“Thank you for taking the time to address my concerns, even if the response to reviewers is now substantially larger than the original manuscript! Overall, I believe the authors have moved a good way to answering my concerns.”*

Response: We would like to thank the Reviewer for a thorough evaluation of our manuscript and for the many constructive criticisms. We are pleased that the Reviewer is largely satisfied with our prior response. We believe the manuscript has greatly benefitted as a result of the helpful comments.

Reviewer 1 Comment #2: *“I have one residual concern - whatever the linear model might say, the growth curves (Figure R1) show significant growth for, minimally, A/J, B6, and DBA mice. I am also concerned about the variance in recorded weights. Are these weights from the study animals used for the turnover study - I don't really see how, since the numbers of animals would then be reducing for each time point, as most individuals are taken? Why no 10d and 14d d for FVB?”*

The variance, and the trends are remarkably noisy, and I would want to see these checked, perhaps on an individual basis. - we do not see such variance in our experience - is there a problem with husbandry? If there was any issue of food availability, then the study would be compromised. The text about animal maintenance should have ensured this variance was more controlled so it remains a curious observation.

However, the biggest issue is the growth of the animals, not the variance in body weight.

Although the sampling regiment is robust to body weight bias, there can be little doubt that in some of these strains, the total pool was expanding - in A/Js for example, by 20% over the experiments duration. Thus the incorporation reflects two terms - turnover and growth. The linear model was not set up to reveal this. I do think this need to be addressed. How much of the variation between strains related to their growth differences? On top of this, the isoP also expands the cardiac pool.

I don't think this should delay publication, but I would prefer a clearer discussion of this issue, and clarification that this is a measurement of one parameter that conceals two processes. To illustrate, Supp Fig 6A refers to 'turnover clusters' ...to my mind, these are 'synthesis clusters'?”

Response: We appreciate the Reviewer's comment on animal growth, and would like to take this opportunity to provide further clarification.

Each time point represents an independent biological replicate group of animals. Therefore, mice studied on day 0 were not the same mice as on day 14. Specifically, day 14 animals were not necessarily older in age at euthanasia, hence body weights do not indicate growth. In this study, we did not perform longitudinal, sequential sampling of identical animals. We apologize if we caused this misunderstanding. As the Reviewer notes that he does not think his comment should delay publication of the manuscript, we have here included additional discussion in manuscript as recommended:

Page 14, lines 19-21 *“Independent groups of 3 mice each were euthanized at 0, 1, 3, 5, 7, 10, 14 days following the first ²H₂O injection of each group at 12:00 noon for sample collection. Labeling began independently for each group.”*

Please also see our response to **Reviewer 1 Comment #3** below for an expanded Discussion section of the manuscript on the parameter quantified in protein turnover measurements.

Reviewer 1 Comment #3: *“Label incorporation (synthesis) is not the same as turnover (the dual processes of synthesis and degradation). Similarly, growth will increase apparent synthesis rates in the absence of degradation. I do hope the authors will be clear in the text about what they are measuring, and how it relates to their findings.*

Thus statements in the text: “utilizes a custom kinetic model to calculate protein turnover rates (k), defined here as the rate (per day) at which a protein pool is proportionally replaced by a combination of synthesis and degradation processes” is incorrect, if the system is expanding (I believe it is, from the growth data).”

Response: We have revised the text to more fully describe the parameter being measured as the Reviewer recommends.

Page 4, lines 9-18 *“...Newly synthesized proteins become labeled with deuterium, whereas degradation of a labeled protein removes deuterium from the protein pool. The changes in proportion of isotopes in the protein pool of each protein species were then measured at 7 time points using a high-resolution Orbitrap mass spectrometer. We developed a computational software, ProTurn, which integrates mass isotopomer signals of all peptides at multiple time points, and utilizes a custom two-compartment kinetics model to calculate protein turnover rates (k)²⁵. We define k here empirically as the measured rate (per day) of label incorporation proportional to the protein pool. Because the protein pool size can change due to physiological or pathological growth as well as differential gene expression, the measure parameter is the outcome of hidden temporal processes including protein synthesis and degradation.”*

Page 12, lines 1-18 *“Large-scale proteome turnover studies have advanced our understanding of health and diseases, and now allow post-transcriptional regulations of individual genes to be detected in a large scale in physiologically relevant in vivo models. Protein turnover measurements have been applied to animal models of cardiac hypertrophy, aging, and caloric restriction, providing unique information on gene regulation in these biological processes. In this study, we assessed protein turnover by modeling percentage label incorporation as a proportion to the total protein pool over time. When the heart protein pool is expanding in size such as during gene up-regulation, physiological growth in animal mass, or pathological growth (hypertrophy), one might assume net protein synthesis rate outpaces degradation rates. The measured parameter therefore reflects the degenerate combinations of net absolute synthesis and degradation rates. Even under non-constant protein pool sizes, we show that two-compartment kinetics effectively models isotope incorporation with no loss in precision, supporting confident relative comparisons between normal and diseased heart for disease signature discovery. Future experiments might combine stable isotope labeling with absolute protein quantification to measure absolute synthesis and degradation rates, which can provide further information on the mechanism of post-transcriptional regulation. Nevertheless, the data presented here already demonstrate the potential to exploit post-transcriptional dynamics to identify disease signatures, and pave the way for further studies to fine-map the mechanisms of gene regulation such as in other disease models and individual cell type studies.”*

Reviewer 1 Comment #4: *Epoxomicin experiment: inhibition of degradation is referred to as decreased turnover (the degradative component) whereas in Supp F2A, turnover refers to label incorporation. Also, if k_d is reduced, why do most proteins decrease in abundance? Was there a full, vehicle-control for this study?*

It's just all a bit confusing - there is no doubt that important behaviours are being seen, but at the same time, it is actually difficult to tease apart mechanistic insights.

Response: Protein abundance data were provided to contrast with changes in the empirical protein turnover rate constant, to demonstrate that the measured turnover rate constants *are responsive to a well-established agent that inhibits protein degradation processes*. Data from our laboratory have shown that cardiac proteasomes are susceptible to inhibition from epoxomicin (Gomes et al. 2009). Functionally, epoxomicin at

0.5 mg/kg per day in mice leads to effective inhibition of proteasomal-dependent protein degradation in vivo (Depre et al. 2006, Jamart et al. 2014, Hedhli et al. 2008).

As the Reviewer and we have noted, the measured parameter reflects label incorporation over the protein pool, which provides independent information over steady-state protein abundance. Proteasomal-dependent protein degradation is but one of multiple protein degradation pathways in the heart. Other protein synthesis and degradation processes are not influenced by epoxomicin which can serve as compensatory mechanisms to decouple protein steady-state abundance. We agree with the Reviewer that the underlying mechanisms of protein turnover are complex and much remains to be learned, which we have noted in the Discussion section of the manuscript.

Reviewer 1 Comment #5: Figure 4: how much of the variance is explained by PC1 and PC2?

Response: We now include the proportional variances of PC1 (53%) and PC2 (27%) in Figure 4.

--

Reviewer #3 (Remarks to the Author):

Reviewer 3 Comment #1: The authors have addressed all questions and concerns of the reviewer.

Response: We thank the Reviewer for their many constructive comments. We believe the manuscript has greatly improved as a result.

--

References

Depre et al. Activation of the Cardiac Proteasome During Pressure Overload Promotes Ventricular Hypertrophy. *Circulation* (2006) 114, 1821.

Gomes et al. Contrasting Proteome Biology and Functional Heterogeneity of the 20 S Proteasome Complexes in Mammalian Tissues. *Molecular & Cellular Proteomics* (2009) 8, 302.

Hedhli et al. Proteasome Inhibition Decreases Cardiac Remodeling after Initiation of Pressure Overload. *AJP Heart and Circulatory Physiology* (2008) 295 4, H1385-H1393.

Jamart et al. Regulation of Ubiquitin-Proteasome and Autophagy Pathways after Acute LPS and Epoxomicin Administration in Mice. *BMC Musculoskeletal Disorders* (2014) 15:166.